# Presentations of children to emergency departments across Europe and the COVID-19 pandemic: A multinational observational study

Ruud G. Nijman[1,2,3]*, Kate Honeyford[4], Ruth Farrugia[5], Katy Rose[1,6], Zsolt Bognar[7], Danilo Buonsenso[8,9], Liviana Da Dalt[10], Tisham De[2], Ian K. Maconochie[1,3], Niccolo Parri[11], Damian Roland[12,13], Tobias Alfven[14], Camille Aupiais[15], Michael Barrett[16,17,18], Romain Basmaci[19], Dorine Borensztajn[20,21], Susana Castanhinha[22], Corinne Vasilico[23], Sheena Durnin[24], Paddy Fitzpatrick[25], Laszlo Fodor[26], Borja Gomez[27,28], Susanne Greber-Platzer[29,30], Romain Guedj[31], Stuart Hartshorn[32,33], Florian Hey[34], Lina Jankauskaite[35], Daniela Kohlfuerst[36], Mojca Kolnik[37], Mark D. Lyttle[38,39], Patrícia Mação[40], Maria Inês Mascarenhas[41], Shrouk Messahel[42], Esra Akyüz Özkan[43], Zanda Pučuka[44], Sofia Reis[45], Alexis Rybak[46,47,48], Malin Ryd Rinder[49], Ozlem Teksam[50], Caner Turan[51], Valtýr Stefánsson Thors[52], Roberto Velasco[53], Silvia Bressan[10], Henriette A. Moll[20], Rianne Oostenbrink[20], Luigi Titomanlio[46,47,48], in association with the REPEM network (Research in European Pediatric Emergency Medicine) as part of the EPISODES study group¶

1 Department of Pediatric Emergency Medicine, Division of Medicine, St. Mary's hospital—Imperial College NHS Healthcare Trust, London, United Kingdom, 2 Faculty of Medicine, Department of Infectious Diseases, Section of Pediatric Infectious Diseases, Imperial College London, London, United Kingdom, 3 Centre for Pediatrics and Child Health, Imperial College London, London, United Kingdom, 4 Faculty of Medicine, School of Public Health, Imperial College London, London, United Kingdom, 5 Department of Child and Adolescent Health, Mater Dei Hospital, Msida, Malta, 6 Division of Emergency Medicine–Pediatrics, University College London NHS Foundation Trust, London, United Kingdom, 7 Department of Pediatric Emergency Medicine, Heim Pal National Pediatric Institute, Budapest, Hungary, 8 Department of Woman and Child Health and Public Health, Fondazione Policlinico Universitario A. Gemelli IRCCS, Rome, Italy, 9 Università Cattolica del Sacro Cuore, Rome, Italy, 10 Division of Pediatric Emergency Medicine, Department of Women's and Children's Health, University Hospital of Padova, Padova, Italy, 11 Emergency Department & Trauma Center, Ospedale Pediatrico Meyer Firenze, Florence, Italy, 12 SAPPHIRE Group, Health Sciences, Leicester University, Leicester, United Kingdom, 13 Pediatric Emergency Medicine Leicester Academic (PEMLA) Group, Leicester Hospitals, Leicester, United Kingdom, 14 Pediatric emergency department, Sachs' Children and Youth Hospital, Stockholm, Sweden, 15 Pediatric Emergency Department, Jean Verdier Hospital, Bondy, France, 16 Pediatric Emergency Department, Children's Health Ireland at Crumlin, Dublin, Ireland, 17 Women's and Children's Health, School of Medicine, University College Dublin, Dublin, Ireland, 18 National Children's Research Centre, Crumlin, Dublin, Ireland, 19 Pediatric Emergency Department, Louis Mourier Hospital, Colombes, France, 20 Department of General Pediatrics, Erasmus MC–Sophia, Rotterdam, the Netherlands, 21 Emergency Department, Medisch Centrum Alkmaar, Noordwest Ziekenhuisgroep, Alkmaar, the Netherlands, 22 Hospital Dona Estefania, Centro Hospitalar de Lisboa Central, Lisbon, Portugal, 23 Department of Pediatrics, Paracelsus Medical University, Salzburg, Austria, 24 Pediatric Emergency Department, Children's Health Ireland at Tallaght, Dublin, Ireland, 25 Pediatric Emergency Department, Children's Health Ireland at Temple Street, Dublin, Ireland, 26 Pediatric Emergency Department, Szent Gyorgy University Teaching Hospital of Fejer County, Szekesfehervar, Hungary, 27 Pediatric emergency department, Cruces University Hospital, Barakaldo, Spain, 28 Biocruces Bizkaia Health Research Institute, Cruces University Hospital, Barakaldo, Spain, 29 Pediatric Emergency Outpatient Clinic, Clinical Division of Pediatric Pulmonology, Allergology and Endocrinology, Department of Pediatrics and Adolescent Medicine, Medical University Vienna, Vienna, Austria, 30 Clinical Division of Pediatric Pulmonology, Allergology and Endocrinology, Department of Pediatrics and Adolescent Medicine, Comprehensive Centre for Pediatrics, Medical University of Vienna, Vienna, Austria, 31 Pediatric Emergency Department, Armand Trousseau Hospital, Paris, France, 32 Pediatric emergency department, Birmingham women's and children's NHS Foundation Trust, Birmingham, United Kingdom, 33 Birmingham Clinical Trials Unit, Institute of Applied Health Research,

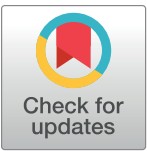

**Data Availability Statement:** All study data are available from via https://doi.org/10.14469/hpc/10685.

**Funding:** RGN was supported by National Institute of Health Research, award number ACL-2018-021-007. The funders had no role in study design, data collection and analysis, decision to publish, or preparation of the manuscript.

**Competing interests:** The authors have declared that no competing interests exist.

**Abbreviations:** CI, confidence interval; COVID-19, Coronavirus Disease 2019; ECDC, European Centre for Disease Prevention and Control; ED, emergency department; EPISODES, Epidemiology, severity and outcomes of children presenting to emergency departments across Europe during the SARS-CoV-2 pandemic; IRR, incidence rate ratio; LRTI, lower respiratory tract infection; MIS-C, Multi Inflammatory Syndrome in Children; PERUKI, Pediatric Emergency Research in the United Kingdom and Ireland; PICU, pediatric intensive care unit; REPEM, Research in European Pediatric Medicine; SARS-CoV-2, Severe Acute Respiratory Syndrome Coronavirus 2.

University of Birmingham, Birmingham, United Kingdom, **34** Pediatric emergency department and pediatric intensive care unit, Dr. von Hauner Children's Hospital, Ludwig-Maximilians-University Munich, Munich, Germany, **35** Hospital of Lithuanian University of Health Sciences Kauno Klinikos, Lithuania, **36** Department of General Pediatrics, Medical University of Graz, Graz, Austria, **37** University Medical Centre Ljubljana, Univerzitetni Klinični Center, Department of Infectious Diseases, Ljubljana, Slovenia, **38** Emergency Department, Bristol Royal Hospital for Children, Bristol, United Kingdom, **39** Faculty of Health and Applied Sciences, University of the West of England, Bristol, United Kingdom, **40** Pediatric Emergency Service, Hospital Pediátrico, Centro Hospitalar e Universitário de Coimbra, Coimbra, Portugal, **41** Departamento da Criança e do Jovem- Urgencia Pediatrica, Hospital Prof. Doutor Fernando da Fonseca, Amadora, Portugal, **42** Pediatric emergency department, Alder Hey Children's NHS Foundation Trust, Liverpool, United Kingdom, **43** Pediatric Emergency Department, Ondokuz Mayıs University, Samsun, Turkey, **44** Pediatric emergency department, Children's Clinical University Hospital, Riga Stradins University, Riga, Latvia, **45** Pediatric Department, Centro Hospitalar Tondela-Viseu, Viseu, Portugal, **46** Pediatric Emergency Department, Hopital Universitaire Robert-Debre, Paris, France, **47** ACTIV, Association Clinique et Thérapeutique Infantile du Val-de-Marne, Créteil, France, **48** INSERM, ECEVE, Université de Paris, Paris, France, **49** Pediatric emergency department, Astrid Lindgrens Children's hospital, Karolinska University, Solna, Sweden, **50** Division of Pediatric Emergency Medicine, Department of Pediatrics, Hacettepe University School of Medicine, Ankara, Turkey, **51** Department of Pediatrics, Division of Emergency Medicine, Mersin City Training and Research Hospital, Toroslar, Mersin, Turkey, **52** Children´s Hospital, Barnaspitali Hringsins, Reykjavik, Iceland, **53** Pediatric emergency unit, Hospital Universitario Río Hortega, Valladolid, Spain

¶ Membership of the EPISODES study group is provided in S1 Appendix
* r.nijman@imperial.ac.uk

# Abstract

## Background

During the initial phase of the Coronavirus Disease 2019 (COVID-19) pandemic, reduced numbers of acutely ill or injured children presented to emergency departments (EDs). Concerns were raised about the potential for delayed and more severe presentations and an increase in diagnoses such as diabetic ketoacidosis and mental health issues. This multinational observational study aimed to study the number of children presenting to EDs across Europe during the early COVID-19 pandemic and factors influencing this and to investigate changes in severity of illness and diagnoses.

## Methods and findings

Routine health data were extracted retrospectively from electronic patient records of children aged 18 years and under, presenting to 38 EDs in 16 European countries for the period January 2018 to May 2020, using predefined and standardized data domains. Observed and predicted numbers of ED attendances were calculated for the period February 2020 to May 2020. Poisson models and incidence rate ratios (IRRs), using predicted counts for each site as offset to adjust for case-mix differences, were used to compare age groups, diagnoses, and outcomes.

Reductions in pediatric ED attendances, hospital admissions, and high triage urgencies were seen in all participating sites. ED attendances were relatively higher in countries with lower SARS-CoV-2 prevalence (IRR 2.26, 95% CI 1.90 to 2.70, $p < 0.001$) and in children aged <12 months (12 to <24 months IRR 0.86, 95% CI 0.84 to 0.89; 2 to <5 years IRR 0.80, 95% CI 0.78 to 0.82; 5 to <12 years IRR 0.68, 95% CI 0.67 to 0.70; 12 to 18 years IRR 0.72, 95% CI 0.70 to 0.74; versus age <12 months as reference group, $p < 0.001$). The lowering

of pediatric intensive care admissions was not as great as that of general admissions (IRR 1.30, 95% CI 1.16 to 1.45, $p < 0.001$). Lower triage urgencies were reduced more than higher triage urgencies (urgent triage IRR 1.10, 95% CI 1.08 to 1.12; emergent and very urgent triage IRR 1.53, 95% CI 1.49 to 1.57; versus nonurgent triage category, $p < 0.001$). Reductions were highest and sustained throughout the study period for children with communicable infectious diseases. The main limitation was the retrospective nature of the study, using routine clinical data from a wide range of European hospitals and health systems.

## Conclusions

Reductions in ED attendances were seen across Europe during the first COVID-19 lockdown period. More severely ill children continued to attend hospital more frequently compared to those with minor injuries and illnesses, although absolute numbers fell.

## Trial registration

ISRCTN91495258 https://www.isrctn.com/ISRCTN91495258.

Author summary

### Why was this study done?

- Reduced numbers of children visiting urgent and emergency care services were reported following the introduction of infection prevention measures during the first phase of the Coronavirus Disease 2019 (COVID-19) pandemic.

- Concerns were raised about potential delays in, and higher acuity of, emergency department (ED) presentations.

### What did the researchers do and find?

- This study compared routine clinical data from children aged 18 years and under presenting to EDs of 38 study sites in 16 European countries between January 2018 until May 2020.

- Reductions in ED attendances were seen for all age groups, with smaller reductions for younger children in some sites.

- More severely ill children continued to attend hospital more frequently compared to those with minor injuries and illnesses, although absolute numbers fell.

### What do these findings mean?

- The findings suggest that the introduction of infection prevention measures can decrease the burden of acute childhood illnesses and injuries.

- There was no clear association of infection prevention measures with an increase in more severe, possibly delayed, presentations.

- For this first phase of the COVID-19 pandemic, the relative increase in cases of diabetic ketoacidosis or mental health issues might have contributed to a biased perception about increased occurrence.

## Introduction

Healthcare systems across Europe continue to be greatly affected by the Coronavirus Disease 2019 (COVID-19) pandemic. Early in the COVID-19 pandemic, urgent and emergency facilities prepared for a potential influx of acutely unwell children and young people [1]. However, evidence emerged that children were less likely to develop symptoms of Severe Acute Respiratory Syndrome Coronavirus 2 (SARS-CoV-2) infection, when compared with adults [2–6]. Moreover, reduced numbers of unwell or injured children visiting urgent and emergency care services were reported, and these seemed to be greatest for children with infectious communicable diseases [7–11]. Typically, these studies did not compare patterns between countries or in relation to different public health strategies.

At the same time, concerns were raised about potential delays in, and higher acuity of, presentations to appropriate healthcare services, as a result of difficulties accessing these services, changes in healthcare provision preferencing virtual consultations, fear of exposure to SARS-CoV-2 in healthcare facilities, and blanket "Stay at Home" statements [12–14]. In the United Kingdom, this resulted in a statement from the Royal Society of Pediatrics and Child Health to reassure parents and caregivers, urging them to seek appropriate urgent and emergency medical attention when worried about the acute illness or injury of their child [15]. Additionally, mostly anecdotal evidence reported increased numbers of specific childhood diagnoses, such as diabetic ketoacidosis [16] and intussusception [17]. These hypothesized a possible link with acute or prior SARS-CoV-2 infection, yet evidence from large-scale cohorts is lacking. Concerns were also raised for the mental health of children resulting from school closures and stay at home orders [18,19].

In this study, we aimed to compare the number of children presenting to emergency departments (EDs) across Europe during the first phase of the COVID-19 pandemic with the 2 previous years; investigating any change in severity of illness and describing the associations with specific diagnoses potentially related to SARS-CoV-2.

## Methods

### Study design, setting, and participants

This retrospective, observational study included 38 sites from 16 European countries as part of the "Epidemiology, severity and outcomes of children presenting to emergency departments across Europe during the SARS-CoV-2 pandemic" (EPISODES) study (trial registration number: ISRCTN91495258) (S1 and S2 Files). Sites were selected from the Research in European Pediatric Medicine (REPEM) and the Pediatric Emergency Research in the United Kingdom and Ireland (PERUKI) networks following the earlier work of Bressan and colleagues [1]. Routine clinical data from all children presenting to the ED were extracted from electronic health records for the period January 1, 2018 to May 17, 2020. The upper age limit varied between

sites at between 16 and 18 years old. This study is reported as per the REporting of studies Conducted using Observational Routinely collected health Data (RECORD) statement (S1 Checklist) and the study protocol is available in the Supporting information (S1 File).

Aggregated, standardized data were uploaded using the REDCap online platform. For the period January 1, 2018 and February 1, 2020, data were collected on a monthly basis. For the period February 2, 2020 to May 17, 2020, on a weekly basis. This amounted to a total of 40 time windows (S1 Table). The clinical report form included 10 different data domains: (1) moment of presentation; (2) patient characteristics; (3) mode of arrival and referral pathway; (4) triage urgency; (5) type of presenting problem and vital signs; (6) diagnostics performed in the ED; (7) treatment in the ED; (8) diagnosis; (9) hospital admission; and (10) duration of ED and hospital stay (S3 File); data availability varied between sites (S1 Fig).

Triage urgency levels, used to determine the urgency of care in the ED, were categorized in 3 predefined categories, defined as emergent-very urgent (or RED-ORANGE, or level 1 to 2), urgent (or YELLOW, or level 3), and standard-nonurgent (or GREEN-BLUE, or level 4 to 5) to allow uniform coding between sites. For diagnosis coding, ICD-10 codes were issued for guidance (S2 Table), but an internally and temporally consistent coding approach was encouraged for each of the individual sites, acknowledging different coding systems and strategies in the ED. This was checked by plotting the diagnoses coding in time as percentage of total number of attendances for each site. To achieve reliable and accurate transformation of local (non-ICD-10) coding systems into the predefined diagnosis categories, training sessions were held and support offered to study sites by the lead investigators. Diagnoses were selected to reflect the broad spectrum of presenting problems to EDs, and their perceived change in incidences during the initial phase of the COVID-19 pandemic, following a consensus methodology among the study steering group. Final selection of diagnoses for analyses, after completion of data quality control process, included (1) common communicable diseases (e.g., tonsillitis, otitis media, lower respiratory tract infection (LRTI), gastrointestinal infection); (2) common minor injuries (e.g., minor head injury, radius fracture); (3) mental health issue; (4) diabetic ketoacidosis; and (5) surgical presentations (e.g., appendicitis, volvulus-intussusception-malrotation, testicular torsion). Severity was defined based on level 1 to 2 urgency classification at triage, any hospital admissions, pediatric intensive care unit (PICU) admission, or death in ED.

## Data analyses

The completeness, quality, and internal consistency of data were checked by plotting the absolute numbers, as well as percentage of total attendances, for each variable of interest in time for the whole study period 2018 to 2020. In order to quantify changes in attendances, we compared observed attendances with predicted numbers of attendances. Predicted numbers of attendances were estimated using monthly data for the 25 months prior to February 3, 2020. As the data had both a trend and seasonal component, we used Holt–Winters exponential smoothing to make short-term monthly forecasts for February, March, April, and May 2020. We adjusted these to weekly estimates of predicted numbers. We plotted predicted ED attendances against the introduction of national infection prevention measures [20]. We also calculated 28-day mean numbers for selected diagnoses, PICU and hospital admission, and death in ED for each month from January through April for the years 2018 to 2020.

We used a Poisson model, adjusted for time since February 3, 2020, to determine if there were differences between age groups, diagnoses, and disposition for patients. For each model, the outcome was the count of attendances per week from the week beginning February 3, 2020 to the week beginning May 4, 2020, with an offset of the predicted number of attendances in

each week. An incidence rate ratio (IRR) >1 indicates higher numbers compared with the reference group, whereas an IRR <1 reflects a higher reduction in numbers. For age groups, the analysis was adjusted for site; for diagnoses and disposition, numbers were too small to make forecasts at site level and we therefore aggregated these across the whole sample. For diagnoses, we completed 2 models, one with 8 separate diagnoses and one where these were divided into 3 groups: surgical presentation (i.e., appendicitis), communicable diseases (i.e., tonsillitis, otitis media, LRTI, and gastroenteritis) and "other" (i.e., mental health issue, radius fracture, and minor head trauma). For 3 diagnosis groups, the number of attendances was too low to make sensible forecasts, namely diabetic ketoacidosis, testicular torsion, and the combined group of intussusception, volvulus, and malrotation. In addition, we determined if there were associations between the change in hospital attendances and the prevalence of SARS-CoV-2 in the country, as per the European Centre for Disease Prevention and Control (ECDC), and the number of COVID-19 measures that were introduced in each hospital in response to the pandemic as previously detailed by Rose and colleagues [21]. Rose and colleagues performed a survey study describing changes in local and regional healthcare pathways, including the diverting of patient groups to or away from the ED, and service provision. The survey covered a total of 37 possible points of change in provision of care for sites without a short stay unit (20 pertaining to service provision and 17 to patient pathways) and 38 possible points of change for those that did (21 service provision and 17 patient pathways).

High-prevalence countries were defined as a cumulative 14-day rate of >80 new cases per 100,000 of the population. For countries with multiple sites, we used an ANOVA to determine if there was evidence that within country differences were greater than between country differences, for total attendances in March and April, adjusted for predicted numbers to account for differences in site sizes. One site (MAL001) was unable to provide information on diagnosis so it was excluded from this section of analysis; 3 sites (SLO001, POR005, and TUR001) did not provide triage data. Two sites were excluded (NL002 and HUN002) from the forecasting analyses and Poisson models as they had missing data in the period before the pandemic (2018). One site (IRE003) was excluded from the Poisson models because it closed to pediatric attendances in response to the pandemic. One site (TUR003) accounted for 18% of all attendances, and we carried out sensitivity analysis to confirm the changes to our findings when including this site. Analyses were performed using R v4.0.0.

## Ethics

Following initial approval by the UK Health Research Authority, all participating sites obtained approval from their national/local institutional review boards (S3 Table). The need for individual patient informed consent was waived. Data sharing agreements were in place.

## Results

### Description of sites, infection prevention measures, and SARS-CoV-2 prevalence

Sites included in the study varied in terms of size and service provision (S4 Table and S2 Fig). The annual number of ED attendances ranged from 4,961 (NL001, 2019) to 295,787 (TUR003, 2019) (S3 Fig). All but 3 sites were tertiary academic hospitals with specialized pediatric EDs; the remaining 3 sites were general teaching hospitals, two of which had dedicated pediatric sections and staff, and one of which had a mixed ED. Sites in Austria, Slovenia, and the Netherlands mainly saw medical presentations, whereas the other sites saw both medical and surgical/trauma presentations. Timing and degree of infection prevention measures were similar

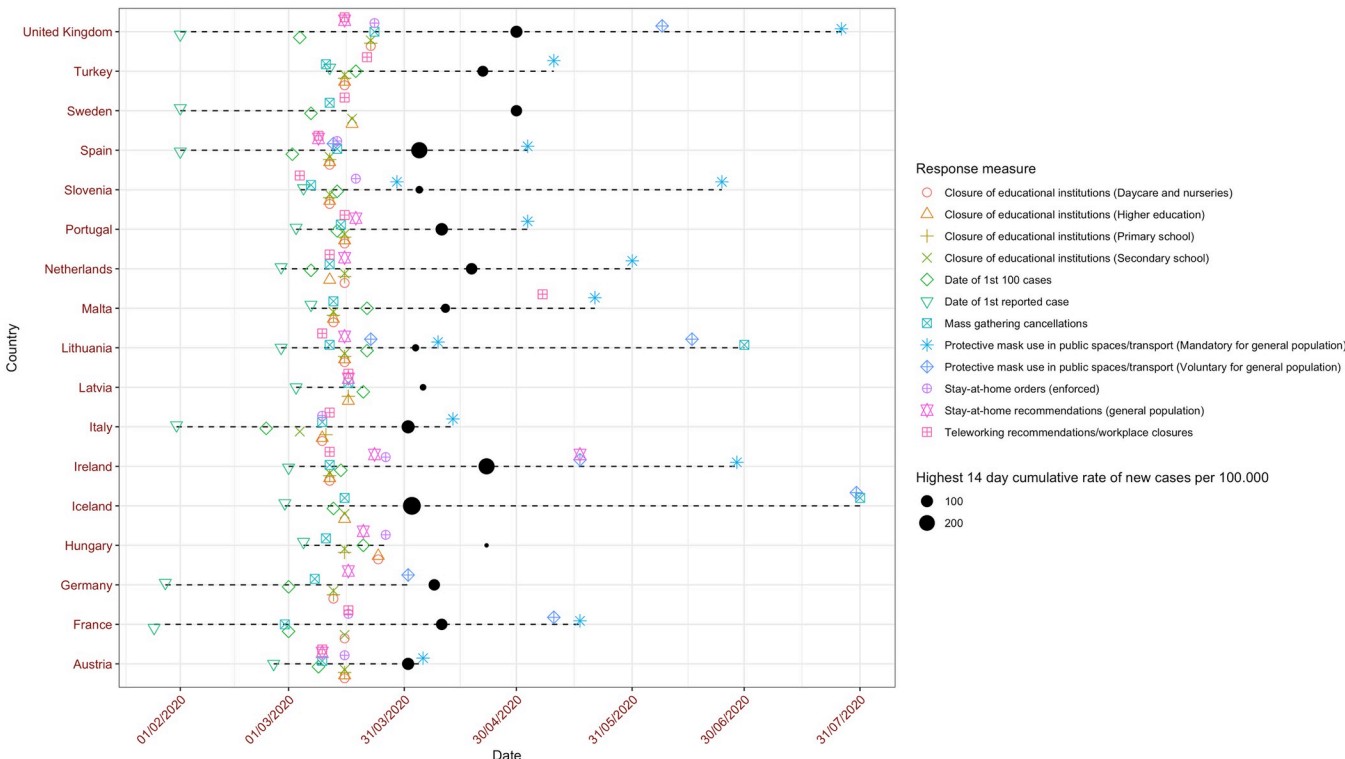

**Fig 1. Timelines of first phase of COVID-19 pandemic in participating countries.** Timelines of the introduction of national infection prevention measures ("Response measures"), as well as dates for the first and first 100 cases of SARS-CoV-2 for each of the countries participating in the EPISODES study. The black circle depicts the date of the highest 14-day cumulative rate of new SARS-CoV-2 cases per 100,000, with the size reflecting the actual case rate. COVID-19, Coronavirus Disease 2019; EPISODES, Epidemiology, severity and outcomes of children presenting to emergency departments across Europe during the SARS-CoV-2 pandemic; SARS-CoV-2, Severe Acute Respiratory Syndrome Coronavirus 2.

across European countries (Fig 1 and S5 Table). Notably, Iceland and Sweden did not close day care, nurseries, or primary education; Germany and the UK kept higher education open; Sweden did not close any public spaces; Hungary and Sweden did not advocate use of face masks; Malta, Iceland, and Sweden did not introduce stay-at-home recommendations; and Germany, Hungary, and Iceland did not formally close workspaces. Highest national prevalence of SARS-CoV-2 varied between countries (Fig 1 and S6 Table).

## Changes in total attendances

All 38 sites had significant reductions in attendances in spring 2020 (Table 1 and Figs 2 and S4). The largest reduction was seen in AUS001 with observed numbers at 5% (95% CI 5% to 6%) of predicted in the week starting March 30, 2020; the smallest peak reduction in ED attendances was at 56% (95% CI 52% to 60%) of predicted in SWE001 during the same week. IRE003 closed for pediatric visits from March 30, 2020 onwards, with most patients diverted to IRE001. Poisson models, adjusted for time since intervention and predicted numbers of attendances, showed that there were significant differences between sites. Observed attendances, with respect to predicted, were relatively higher in sites in France, Sweden, Ireland, Iceland, Latvia, and the Netherlands, where observed attendance rates were greater than 50% of predicted. However, there was considerable overlap between all sites when 95% confidence intervals were considered. Results of the Poisson models suggest that attendances in Spring 2020 were higher in EDs in countries with lower SARS-CoV-2 prevalence (IRR 2.26, 95% CI

**Table 1. Summary data on the lowest observed number of ED attendances during COVID-19 for each participating center.**

| Site | Week of first public health measure | Week of highest reduction | Observed number of ED attendances | Total % of predicted numbers (95% CI) | Overall reduction$ | Date of highest 14-day cumulative rate of new cases per 100,000 | Highest cumulative 14-day rate of new cases per 100,000 | Total changes to health system | Total changes as % of possible changes |
|---|---|---|---|---|---|---|---|---|---|
| Austria | 09-03-2020 | | | | 75% | 02-04-2020 | 102.33 | | |
| AUS001 | | 13-04-2020 | 16 | 5% (5% - 6%)* | 85% | | | 5 | 13.5% |
| AUS003 | | 30-03-2020 | 119 | 17% (15% - 18%) | 72% | | | 1 | 2.7% |
| AUS004 | | 30-03-2020 | 114 | 21% (19% - 23%) | 71% | | | 0 | 0.0% |
| France | 24-02-2020 | | | | 18% | 11-04-2020 | 86.12 | | |
| FR001 | | 30-03-2020 | 470 | 41% (35% - 48%) | 28% | | | 2 | 5.3% |
| FR002 | | 30-03-2020 | 154 | 52% (40% - 77%) | -6% | | | 1 | 2.6% |
| FR003 | | 30-03-2020 | 285 | 42% (36% - 49%) | 28% | | | 0 | 0.0% |
| FR004 | | 13-04-2020 | 115 | 34% (27% - 48%) | 23% | | | 0 | 0.0% |
| Germany | 02-03-2020 | | | | | 09-04-2020 | 86.36 | | |
| GER001 | | 30-03-2020 | 165 | 36% (33% - 39%) | 53% | | | 0 | 0.0% |
| Hungary | 09-03-2020 | | | | | 23-04-2020 | 13.34 | | |
| HUN001 | | 23-03-2020 | 279 | 35% (33% - 37%) | 52% | | | 4 | 10.5% |
| Iceland | 16-03-2020 | | | | | 03-04-2020 | 277.04 | | |
| ICE001 | | 06-04-2020 | 132 | 49% (44% - 55%) | 40% | | | 3 | 7.9% |
| Ireland | 09-03-2020 | | | | 56% | 23-04-2020 | 213.02 | | |
| IRE001 | | 23-03-2020 | 404 | 50% (46% - 55%) | 27% | | | 3 | 7.9% |
| IRE002 | | 30-03-2020 | 333 | 32% (28% - 36%) | 55% | | | 4 | 10.5% |
| IRE003 | | 30-03-2020 | 0 | 0%+ | 86% | | | 13 | 34.2% |
| Italy | 02-03-2020 | | | | 70% | 02-04-2020 | 124.03 | | |
| IT001 | | 23-03-2020 | 105 | 20% (20% - 21%) | 67% | | | 1 | 2.6% |
| IT002 | | 23-03-2020 | 36 | 12% (11% - 13%) | 73% | | | 8 | 21.1% |
| Latvia | 16-03-2020 | | | | | 06-04-2020 | 20.52 | | |
| LAT001 | | 30-03-2020 | 491 | 38% (37% - 39%) | 53% | | | 0 | 0.0% |
| Lithuania | 09-03-2020 | | | | | 04-04-2020 | 25.12 | | |
| LIT001 | | 30-03-2020 | 164 | 26% (24% - 27%) | 62% | | | 9 | 23.7% |
| Malta | 09-03-2020 | | | | | 12-04-2020 | 46.20 | | |
| MAL001 | | 30-03-2020 | 68 | 13% (12% - 15%) | 80% | | | 1 | 2.7% |
| The Netherlands | 09-03-2020 | | | | | 19-04-2020 | 86.57 | | |
| NL001 | | 16-03-2020 | 38 | 36% (33% - 38%) | 44% | | | 1 | 2.7% |
| Portugal | 09-03-2020 | | | | 71% | 11-04-2020 | 109.02 | | |
| POR001 | | 13-04-2020 | 305 | 25% (23% - 28%) | 70% | | | 1 | 2.6% |
| POR003 | | 30-03-2020 | 365 | 22% (20% - 24%) | 70% | | | 2 | 5.3% |
| POR004 | | 30-03-2020 | 132 | 13% (12% - 15%) | 74% | | | 6 | 15.8% |
| POR005 | | 20-04-2020 | 117 | 21% (18% - 23%) | 69% | | | 1 | 2.6% |
| Slovenia | 02-03-2020 | | | | | 05-04-2020 | 28.55 | | |
| SLO001 | | 30-03-2020 | 12 | 8% (7% - 8%) | 74% | | | 3 | 7.9% |
| Spain | 09-03-2020 | | | | 70% | 05-04-2020 | 217.56 | | |
| SP001 | | 23-03-2020 | 256 | 26% (24% - 28%) | 62% | | | 1 | 2.6% |
| SP002 | | 30-03-2020 | 51 | 11% (10% - 11%) | 77% | | | 0 | 0.0% |
| Sweden | 09-03-2020 | | | | 36% | 01-05-2020 | 83.60 | | |
| SWE001 | | 30-03-2020 | 661 | 56% (52% - 60%)^ | 31% | | | 0 | 0.0% |
| SWE002 | | 06-04-2020 | 253 | 48% (44% - 52%) | 40% | | | 0 | 0.0% |
| Turkey | 09-03-2020 | | | | 68% | 22-04-2020 | 74.97 | | |

*(Continued)*

**Table 1.** (Continued)

| Site | Week of first public health measure | Week of highest reduction | Observed number of ED attendances | Total % of predicted numbers (95% CI) | Overall reduction$ | Date of highest 14-day cumulative rate of new cases per 100,000 | Highest cumulative 14-day rate of new cases per 100,000 | Total changes to health system | Total changes as % of possible changes |
|---|---|---|---|---|---|---|---|---|---|
| TUR001 | | 27-04-2020 | 98 | 33% (29% - 38%) | 59% | | | 3 | 7.9% |
| TUR002 | | 06-04-2020 | 233 | 15% (14% - 17%) | 72% | | | 6 | 15.8% |
| TUR003 | | 13-04-2020 | 882 | 14% (13% - 15%) | 75% | | | 5 | 13.5% |
| United Kingdom | 16-03-2020 | | | | 63% | 01-05-2020 | 99.25 | | |
| UK001 | | 30-03-2020 | 387 | 30% (28% - 32%) | 63% | | | 3 | 7.9% |
| UK002 | | 30-03-2020 | 72 | 18% (17% - 21%) | 71% | | | 3 | 7.9% |
| UK004 | | 23-03-2020 | 487 | 36% (34% - 39%) | 55% | | | 2 | 5.3% |
| UK005 | | 13-04-2020 | 76 | 14% (12% - 17%) | 71% | | | 1 | 2.6% |
| UK006 | | 30-03-2020 | 401 | 33% (31% - 35%) | 54% | | | 2 | 5.3% |

The starting date of the week where the observed numbers of ED attendances had the largest difference from predicted numbers expressed in % with 95% confidence intervals. With

* AUS001 having the largest reduction from predicted numbers and ^ SWE001 the least change from predicted numbers. In addition, highest national SARS-CoV-2 infection rates are given for each of the study sites, as reported by the ECDC [20],with a threshold of 80 cases per 100,000 to indicate low (green) and high (orange) prevalence. Total changes to health system as per Rose and colleagues [21].

$Overall reduction calculated from introduction of first national public health measure until end of study period. If multiple sites per country, overall reduction for the country was estimated using the average of the overall reductions of each individual site.

+IRE003 was closed for pediatric attendances from March 30, 2020 onwards.

HUN002, NL002 excluded from table owing to missing data in 2018.

COVID-19, Coronavirus Disease 2019; ECDC, European Centre for Disease Prevention and Control; ED, emergency department; SARS-CoV-2, Severe Acute Respiratory Syndrome Coronavirus 2.

1.90 to 2.70, $P < 0.001$) (Table 2). We found a relationship between the number of introduced organizational COVID-19 measures and ED attendances and more organizational COVID-19 measures were associated with lower numbers of ED attendances when adjusted for predicted ED attendances (IRR 0.13, 95% CI 0.11 to 0.16, when sites with 4 or more measures were compared to sites with no measures, $P < 0.001$). Similarly, larger reductions in ED attendances were seen in mixed adult and pediatric academic hospitals (versus standalone children's hospital, IRR 3.49, 95% CI 2.89 to 4.24, $P < 0.001$; general nonuniversity hospital, IRR 2.73, 95% CI 2.28 to 3.30, $P < 0.001$) and urban hospitals (versus mixed urban and rural hospitals, IRR 5.33, 95% CI 4.44 to 6.46, $P < 0.001$). ED attendances across all age groups significantly reduced (S5 and S6 Figs). Attendances in children aged above 12 months were reduced more than children below 12 months (12 to <24 months IRR 0.86, 95% CI 0.84 to 0.89; 2 to <5 years IRR 0.80, 95% CI 0.78 to 0.82; 5 to <12 years IRR 0.68, 95% CI 0.67 to 0.70; 12 to 18 years IRR 0.72, 95% CI 0.70 to 0.74; versus age <12 months as reference group, all $P < 0.001$) (Table 2). There was insufficient evidence to conclude that this pattern continued with increasing age for children aged 12 months and older. Patterns between sites within the same country appeared similar (S7 Fig) with strong evidence that between country differences were greater than within country differences (F value: 6.453; $p$: 0.002).

## Triage urgency

Overall, there was a higher reduction (observed compared to predicted) in children with lower triage urgency when compared to children with higher triage classification (urgent triage, IRR

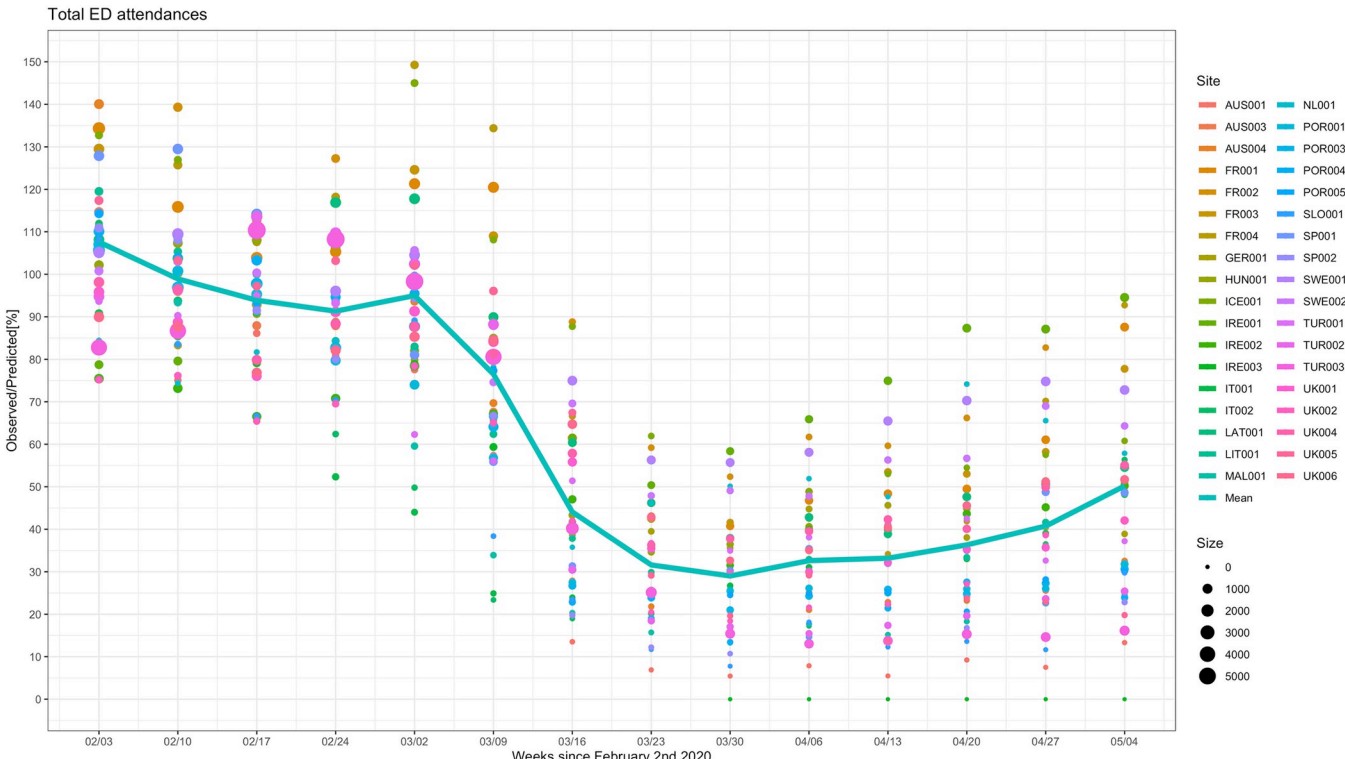

**Fig 2. Observed versus predicted ED attendances (%).** The observed versus predicted number of children presenting to EDs in countries across Europe in the weeks following February 2, 2020 until May 11, 2020, for all sites combined. The color and the size of the dots reflect the actual number of ED attendances for each site and for each time window. The line connects the mean of the observed vs. predicted point estimates for each of the individual sites for each time window.

1.10, 95% CI 1.08 to 1.12, $P < 0.001$; emergent and very urgent triage IRR 1.53, 95% CI 1.49 to 1.57, $P < 0.001$; versus nonurgent triage category), even though clear reductions were seen for all triage categories (S8 Fig).

## Hospital and PICU admissions

Hospital and PICU admissions were fewer than predicted (Figs 3, 4 and S9). We did not observe an increase in the number of deaths in ED (IRR 1.75, 95% CI 0.88 to 3.07, $p = 0.08$). The change in PICU admissions (IRR 1.30, 95% CI 1.16 to 1.45, versus general admissions) was not as great as the change in general admissions.

## Diagnoses

The 28-day mean numbers for common communicable diseases decreased in absolute and relative frequencies (Table 3 and Fig 5A), in particular for tonsillitis, otitis media, gastrointestinal infections, and LRTIs. Decreases were also seen in common childhood injuries such as minor head injuries and radius fractures (Fig 5B). No increase in absolute numbers were seen for several uncommon diagnoses suggested to be linked with SARS-CoV-2 infection, such as diabetic ketoacidosis (Fig 5C), intussusception, and testicular torsion (Fig 5B), even when stratified for high-SARS-CoV-2 prevalence countries (S10 Fig). Mental health attendances declined during the first phase of the COVID-19 pandemic in absolute terms, but this corresponded with an increase in relative frequency (Fig 5C). Fig 6, reflecting the observed versus predicted numbers for the 8 selected diagnoses, shows that the change of children and young people with

**Table 2. Poisson regression models for ED attendances.**

| | IRR (95% Confidence Interval) | p-value |
|---|---|---|
| **Number of COVID-19 measures in hospital**[$] | | |
| No measures–reference group | | |
| 1 measure | 0.426 (0.391 - 0.463) | <0.001 |
| 2 measures | 0.760 (0.717 - 0.806) | <0.001 |
| 3 measures | 0.454 (0.413 - 0.499) | <0.001 |
| 4+ measures | 0.130 (0.108 - 0.155) | <0.001 |
| **SARS-CoV-2 prevalence**[^] | | |
| Low prevalence | 2.256 (1.904 - 2.700) | <0.001 |
| **Type of hospital** | | |
| Mixed tertiary hospital–reference group | | |
| Standalone tertiary children's hospital | 3.490 (2.894 - 4.241) | <0.001 |
| General nonuniversity teaching hospital | 2.733 (2.280 - 3.303) | <0.001 |
| Urban–reference group | | |
| Urban and rural mixed | 5.333 (4.439 - 6.460) | <0.001 |
| **Age group** | | |
| 0-<12 months–reference group | | |
| 12-<24 months | 0.862 (0.838 - 0.886) | <0.001 |
| 2-<5 years | 0.799 (0.780 - 0.819) | <0.001 |
| 5-<12 years | 0.682 (0.666 - 0.698) | <0.001 |
| 12–18 years | 0.719 (0.698 - 0.739) | <0.001 |
| **Triage urgency classification** | | |
| Nonurgent and standard triage categories–reference group | | |
| Urgent | 1.096 (1.076 - 1.116) | <0.001 |
| Emergent and very urgent | 1.530 (1.488 - 1.573) | <0.001 |
| **Diagnosis I** | | |
| Appendicitis–reference group | | |
| Gastro-intestinal infections | 0.279 (0.253 - 0.308) | <0.001 |
| Minor head injury | 0.783 (0.709 - 0.866) | <0.001 |
| LRTI | 0.357 (0.323 - 0.396) | <0.001 |
| Mental health issues | 0.688 (0.609 - 0.777) | <0.001 |
| Otitis media | 0.231 (0.206 - 0.260) | <0.001 |
| Radius fracture | 0.732 (0.654 - 0.819) | <0.001 |
| Tonsillitis | 0.189 (0.172 - 0.208) | <0.001 |
| **Diagnosis II** | | |
| Surgical presentation–appendicitis–reference group | | |
| Communicable diseases | 0.238 (0.253 - 0.308) | <0.001 |
| Other | 0.754 (0.709 - 0.866) | <0.001 |
| **Outcome** | | |
| Admission–reference group | | |
| Death | 1.749 (0.876 - 3.069) | 0.077 |
| PICU Admission | 1.295 (1.157 - 1.445) | <0.001 |

To derive IRRs, the predicted counts for each individual site were used as an offset in the Poisson model to account for case-mix differences between sites.

[$]The number of changes made in each hospital in response to the pandemic as previously detailed by Rose and colleagues [21].

[^]Low-prevalence countries were defined as a cumulative 14-day rate of < = 80 new cases per 100,000 of the population as per the ECDC [20].

COVID-19, Coronavirus Disease 2019; ECDC, European Centre for Disease Prevention and Control; ED, emergency department; IRR, incidence rate ratio; LRTI, lower respiratory tract infection; PICU, pediatric intensive care unit; SARS-CoV-2, Severe Acute Respiratory Syndrome Coronavirus 2.

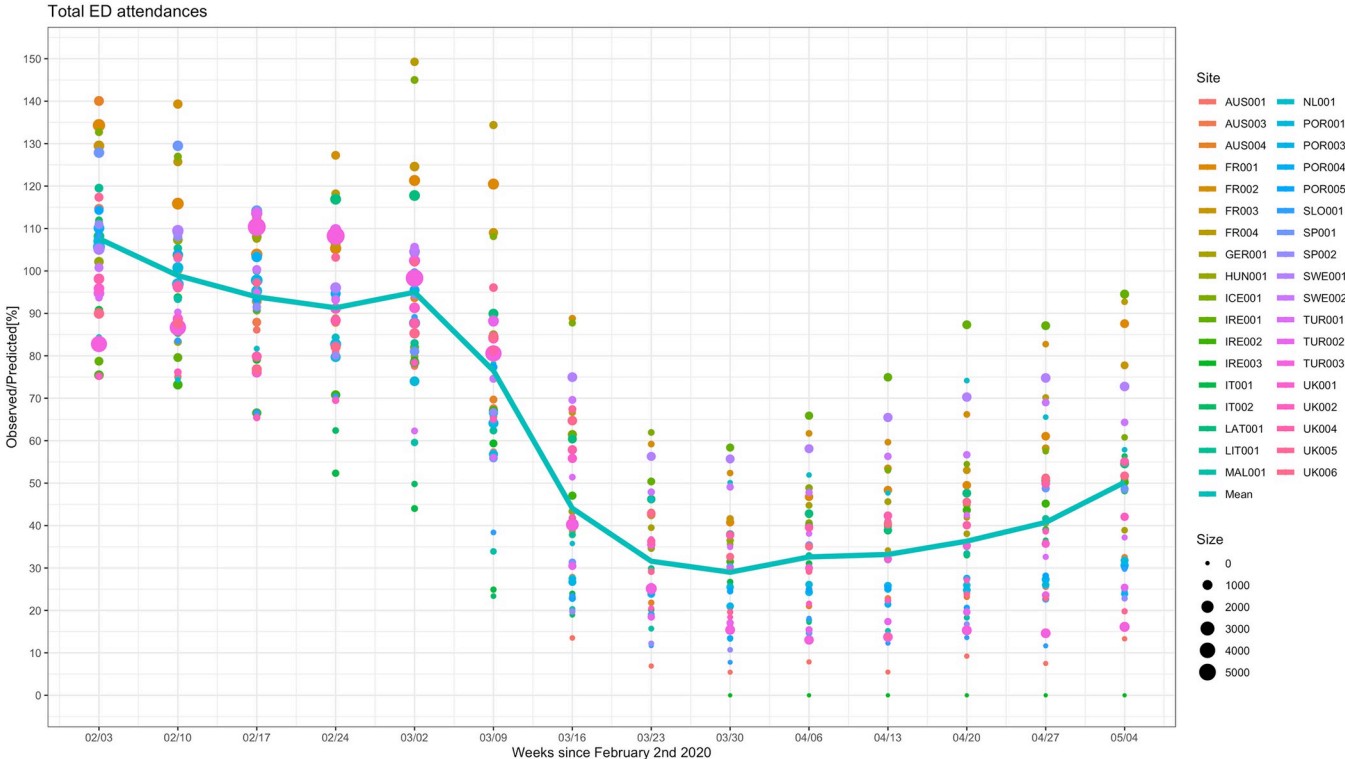

**Fig 3. Observed versus predicted hospital admissions for patients attending the ED (%).** The observed versus predicted number of children admitted to hospital from the ED in countries across Europe in the weeks following February 2, 2020 until May 11, 2020, for all sites combined. The color and the size of the dots reflect the actual number of ED attendances for each site and for each time window. The line connects the mean of the observed vs. predicted point estimates for each of the individual sites for each time window.

appendicitis was less than for the other diagnoses groups. Mental health issues, radius fractures, and minor head injuries were all affected, but there was evidence that attendances increased from the end of March. In contrast, attendances for LRTI, otitis media, gastrointestinal infections, and tonsillitis remained low. Poisson models showed no significant difference between mental health, minor head trauma, and radius fracture. There was evidence of significant difference between infections and trauma and mental health, with bigger reductions in infections. When communicable diseases were combined, there was a clear difference between surgical presentation (appendicitis), communicable diseases, and "other" (mental health, radius fracture, and head trauma) (Table 2).

## Sensitivity analyses

The sensitivity analyses for the Poisson modelling without TUR003 resulted in IRRs slightly nearer to one, meaning all associations were slightly weaker (S7 Table). The change of the coefficient for tonsillitis was notable, increasing the IRR from 0.19 (95% CI 0.17 to 0.21) to 0.37 (95% CI 0.34 to 0.41). There was also a change between PICU admissions (IRR 1.13, 95% CI 1.00 to 1.28, $p = 0.045$) and admissions in general. Though the comparison remained statistically significant, the association was weaker.

## Discussion

Reductions in the numbers of children attending EDs were consistently seen across Europe during the first phase of the COVID-19 pandemic. There was variation between countries, but

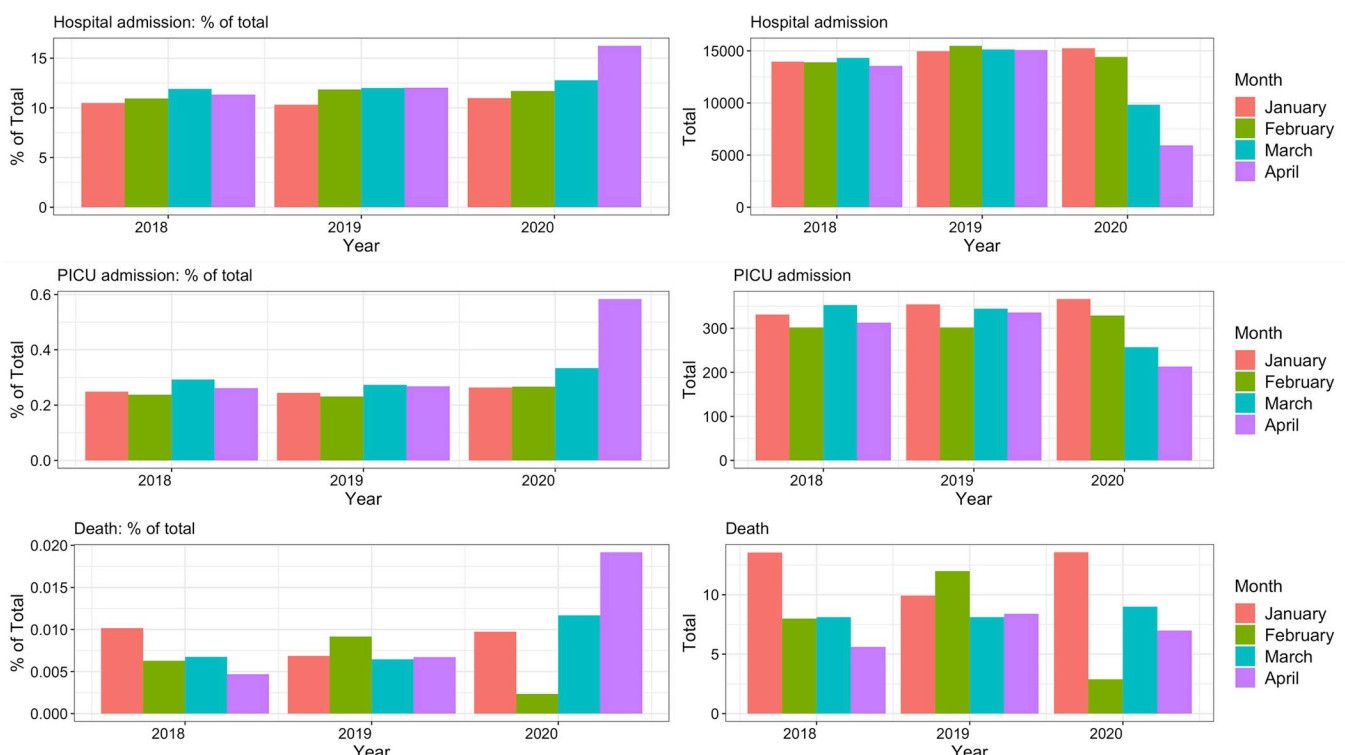

**Fig 4. Hospital admissions, intensive care admissions, and deaths in the ED for the period January–April over a 3-year period.** Percentages of total ED attendances (left) and absolute numbers (right) of children admitted to hospital (top), PICUs (middle), or died in the ED (bottom); comparing the 28-day mean numbers for the months of January–April for 2018 vs. 2019 vs. 2020. ED, emergency department; PICU, pediatric intensive care unit.

within countries patterns were similar. The levels to which ED attendances decreased appeared to be related to the introduction and extend of infection prevention measures, changes made to local health systems, type of hospital, and national SARS-CoV-2 prevalence. Attendances were relatively higher in some sites with fewer or less strict national infection prevention measures (e.g., Sweden, Iceland), but this was not true for others (e.g., France). ED attendances were seen for all age groups, with smaller reductions in children aged below 1 year. The reduction in numbers was largest and sustained for communicable diseases, whereas other groups of diagnoses trended towards normal levels of ED attendances by the end of the study period after initial reduced ED attendance rates.

Our findings of reduced pediatric ED attendances are consistent with other studies from around the world [7–11,22–24]. The observed reduction in ED attendances will likely be multifactorial, including changed parental health-seeking behavior, modified and newly introduced healthcare pathways, and fewer circulating and reduced transmission of infectious pathogens. For example, children with asthma often frequent EDs, but they had fewer exacerbations needing ED visits during the first phase of the COVID-19 pandemic. Proposed reasons include reduced air pollution, reduced social mixing with exposures to viral trigger, and improved compliance with medication at home [25,26]. All this appeared to have affected presentations of children and young people with a low triage urgency and with minor injuries and illnesses most.

Earlier studies suggested that infection prevention measures may have resulted in delayed presentations to hospitals [12,13,27–29]. In our study, children with more severe conditions, as measured by triage urgency, need for hospital admission, and PICU and death, continued

**Table 3. Totals of selected clinical diagnoses in the ED for the period January–April over a 3-year period.**

| | 2018 | | | | 2019 | | | | 2020 | | | |
|---|---|---|---|---|---|---|---|---|---|---|---|---|
| | January | February | March | April | January | February | March | April | January | February | March | April |
| Tonsillitis | 17,627 (12.9%) | 16,781 (12.9%) | 15,249 (12.4%) | 14,722 (12.0%) | 24,939 (16.8%) | 16,231 (12.1%) | 15,898 (12.3%) | 18,751 (14.6%) | 26,118 (18.4%) | 17,668 (14.0%) | 10,639 (13.5%) | 2,382 (6.3%) |
| Otitis media | 4,917 (3.6%) | 4,814 (3.7%) | 4,063 (3.3%) | 3,639 (3.0%) | 4,843 (3.3%) | 4,370 (3.2%) | 3,900 (3.0%) | 3,719 (2.9%) | 4,115 (2.9%) | 4,452 (3.5%) | 2,156 (2.7%) | 407 (1.1%) |
| LRTIs | 9,129 (6.7%) | 8,981 (6.9%) | 7,197 (5.8%) | 5,407 (4.4%) | 10,684 (7.2%) | 10,294 (7.7%) | 6,519 (5.1%) | 5,415 (4.2%) | 9,374 (6.6%) | 9,028 (7.2%) | 5,667 (7.2%) | 1,059 (2.8%) |
| Gastrointestinal infections | 7,809 (5.7%) | 8,488 (6.5%) | 8,946 (7.3%) | 9,103 (7.4%) | 10,313 (6.9%) | 9,949 (7.4%) | 11,345 (8.8%) | 11,066 (8.6%) | 9,158 (6.4%) | 8,726 (6.9%) | 4,719 (6.0%) | 1,905 (5.1%) |
| Appendicitis | 399 (0.3%) | 412 (0.3%) | 434 (0.4%) | 424 (0.3%) | 455 (0.3%) | 437 (0.3%) | 505 (0.4%) | 443 (0.3%) | 431 (0.3%) | 435 (0.3%) | 332 (0.4%) | 307 (0.8%) |
| Testicular torsion | 119 (0.1%) | 118 (0.1%) | 124 (0.1%) | 117 (0.1%) | 143 (0.1%) | 111 (0.1%) | 147 (0.1%) | 136 (0.1%) | 125 (0.1%) | 125 (0.1%) | 99 (0.1%) | 103 (0.3%) |
| Intussusception, volvulus, and malrotation | 33 (0.0%) | 37 (0.0%) | 56 (0.0%) | 58 (0.0%) | 40 (0.0%) | 58 (0.0%) | 63 (0.0%) | 60 (0.0%) | 55 (0.0%) | 44 (0.0%) | 53 (0.1%) | 30 (0.1%) |
| Mental health conditions | 705 (0.5%) | 652 (0.5%) | 718 (0.6%) | 736 (0.6%) | 811 (0.5%) | 821 (0.6%) | 853 (0.7%) | 757 (0.6%) | 791 (0.6%) | 802 (0.6%) | 603 (0.8%) | 388 (1.0%) |
| Diabetic ketoacidosis | 61 (0.0%) | 60 (0.0%) | 56 (0.0%) | 64 (0.1%) | 68 (0.0%) | 54 (0.0%) | 61 (0.0%) | 62 (0.0%) | 62 (0.0%) | 64 (0.1%) | 55 (0.1%) | 50 (0.1%) |
| Radius fracture | 783 (0.6%) | 891 (0.7%) | 853 (0.7%) | 1,311 (1.1%) | 886 (0.6%) | 1,036 (0.8%) | 1,153 (0.9%) | 1,362 (1.1%) | 809 (0.6%) | 935 (0.7%) | 730 (0.9%) | 592 (1.6%) |
| Minor head injury | 2,709 (2.0%) | 2,976 (2.3%) | 2,985 (2.4%) | 3,392 (2.8%) | 2,682 (1.8%) | 2,730 (2.0%) | 3,043 (2.4%) | 3,169 (2.5%) | 2,372 (1.7%) | 2,415 (1.9%) | 1,728 (2.2%) | 1,472 (3.9%) |

Absolute numbers (and % of total children seen in the ED) of children with diagnosis of (a) tonsillitis; (b) otitis media; (c) LRTIs; (d) gastrointestinal infections; (e) appendicitis; (f) testicular torsion; (g) intussusception, volvulus, and malrotation; (h) mental health issues; (i) diabetic ketoacidosis; (j) radius fracture; and (k) minor head injury; comparing the 28-day mean numbers for the months of January–April for 2018 vs. 2019 vs. 2020.

ED, emergency department; LRTI, lower respiratory tract infection.

to attend hospital more frequently compared to those with minor injuries and illnesses, although overall absolute numbers fell. This was in line with other studies reporting similar reductions in children with high triage urgency or need for hospital admission [7,30–38].

Defining the harm of delayed presentations, as well as establishing what contributed to a possible delay, can be difficult [39]. In an attempt to distinguish the delay in seeking care from harm sustained, Roland and colleagues concluded that only a minority (6 out of 51 (11.8%)) of children with a potential delay in presentation were admitted to 1 of 7 hospitals [40]. Contradictory conclusions have been reported for the delay in presentations and for potential harm sustained for diagnoses of appendicitis [41,42] and testicular torsion [43–46], portraying a picture that organizing regional healthcare delivery is important to ensure continued access to pediatric urgent and emergency care during a pandemic. In addition, our data showed that, despite overall falling ED attendances, presentations requiring surgical interventions remained stable, reiterating that access to surgical teams and the ability to perform emergent surgical procedures are crucial.

Evidence is mounting that SARS-CoV-2 is directly involved in the pathogenesis of new onset diabetes [47,48]. Unsworth and colleagues first reported an increase of new onset type 1 diabetes in children and a possible link with SARS-CoV-2 in the UK [16]. Additional cohort studies found divergent associations between SARS-CoV-2, new onset diabetes, and decompensation of preexisting diabetes [49–52]. Our data did not identify increased incidence of

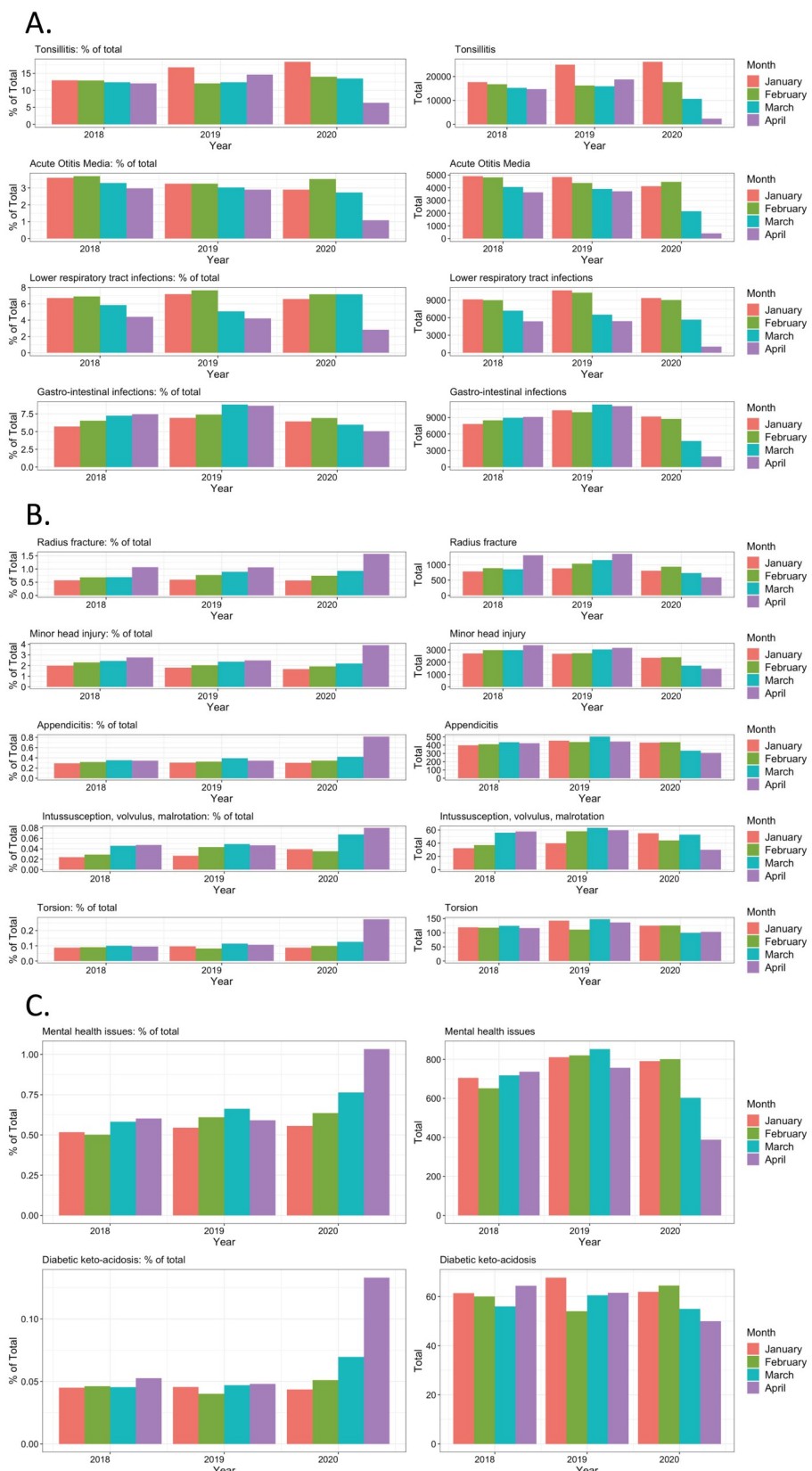

**Fig 5. Selected clinical diagnoses in the ED for the period January–April over a 3-year period.** Percentages of total ED attendances (left) and absolute numbers (right) of children with diagnosis of (A) common communicable diseases (tonsillitis, otitis media, LRTI, GI infections), (B) minor injuries and surgical presentations (radius fracture, minor head injury, appendicitis, intussusception, volvulus and malrotation (combined group), testicular torsion,), and (C) mental health issues and diabetic ketoacidosis; comparing the 28-day mean numbers for the months of January–April for 2018 vs. 2019 vs. 2020. ED, emergency department; GI, gastrointestinal; LRTI, lower respiratory tract infection.

diabetic ketoacidosis during the first phase of the COVID-19 pandemic. It might well be that clusters of new onset diabetes can be found in high-prevalence areas and that we failed to capture this in our study. Likewise, if SARS-CoV-2 acts as a precipitator, there might be a delay in the manifestation of new onset diabetes, and with reduced prevalence of typical viral triggers, this increase might only become apparent later in the pandemic [53]. We were not able to differentiate between new onset diabetes and decompensation of preexisting diabetes.

We found a reduction of children with mental health conditions presenting to the EDs during the first phase of the COVID-19 pandemic in Europe, similar to findings from studies elsewhere [9,54–57]. This is unlikely to reflect the considerable mental health issues encountered in the wider pediatric and adolescent populations [58] and of the experiences later in the pandemic, with, among others, reported increases in eating disorders in children and young people [59]. Joyce and colleagues observed an overall decrease in mental health issues in ED, albeit an increase in self-harm and deliberate ingestions [60]. Despite the reduction in absolute numbers, there was an increase in the proportion of attendances attributable to mental health potentially contributing to the heightened awareness for mental health issues in the first COVID-19 wave.

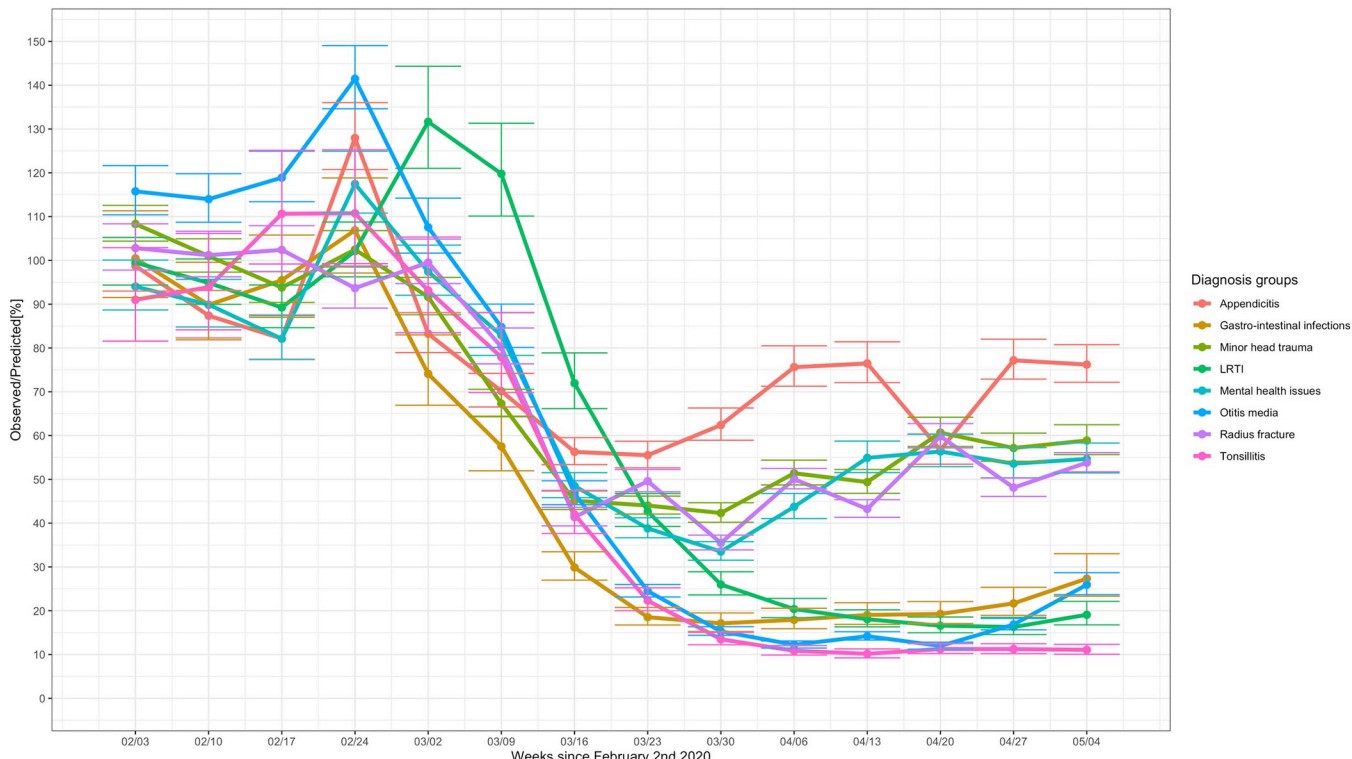

**Fig 6. Observed versus predicted number of selected diagnoses (%).** The observed versus predicted numbers of 8 selected diagnoses for all sites combined, for the period following February 2, 2020 until May 4, 2020. The error bars indicate the 80% prediction intervals.

Prior to the current COVID-19 pandemic, limited data were available describing the association of infection prevention measures on urgent and emergency pediatric care in high-income countries. One study found a decrease in respiratory infections of 42% and decreased ED attendances of 28% following school closures for an influenza outbreak in Israel [61]. Similar patterns were seen during the SARS outbreak in 2003 [62–64] and the MERS outbreak in 2015 [65]. These studies also reported a larger reduction in ED utilization for children than for adults. In contrast, the 2009 H1N1 influenza pandemic generally led to increased ED utilization, with higher levels of acuity [66–68]. One previous study had reported reduced pediatric ED attendance rates for flu-like illness and respiratory tract infections following school closures [69]. Another study reported increased pediatric ED attendance numbers following media reports on health threats of the H1N1 virus [70]. Altogether, previous evidence of infectious disease outbreaks suggests a similar impact on pediatric urgent and emergency care following the introduction of public health and infection prevention measures. However, this is to a lesser extent than what was observed with the COVID-19 pandemic and one that is dependent on childhood susceptibility for the infectious pathogen.

### Strengths and limitations

Our study presents multinational data enabling the comparison between infection prevention measures, national SARS-CoV-2 prevalence, and the association with acute illness and injuries in children between European countries. Most participating sites were tertiary institutions, with dedicated pediatric emergency medicine teams, with potential implications for the generalizability of our findings. At present, no standardized data extraction system for pediatric urgent and emergency care exists between European countries; and the EPISODES study is, to our knowledge, the first to navigate the difficulties of dealing with different data systems, data availability, and varying coding practices. Hence, also limited by the time restrictions caused by the COVID-19 pandemic, some sites were not able to provide data for all domains, and 2 sites (NL002 and HUN002) were only able to provide data for part of the study duration.

Limitations of electronic health records to describe patients' diagnoses are well known [71]. Some of the participating study sites had unique non-ICD-10-based coding systems, and we urged all study teams to be consistent in transforming local data to fit the study clinical report form and we implemented a rigorous data quality process to ensure validity of coding in time. Although most diagnoses linked to SARS-CoV-2 in children were included in the predefined clinical report form, other diagnoses might be of interest in future studies. Of note, coding for children with Multi Inflammatory Syndrome in Children (MIS-C) proved unreliable, with no unique diagnostic codes available for this new disease in automated coding systems.

As the data were collected in aggregated form, thereby negating some of the difficulties with data protection regulations, we were not able to stratify for severity of specific diagnosis or age groups. This also introduced risks of overstratification during analyses. We observed large differences between sites for the number of annual ED attendances and the number of patients with high triage urgency and hospital admissions, reflecting both case-mix and diversity of patient management. We analyzed data mostly on a site-by-site basis, by using predicted versus observed ratios, and thus dealing with heterogeneity between sites. In addition, although very few study sites restructured local healthcare pathways diverting urgent and emergency care to alternative healthcare facilities, this does not fully rule out changes to access to healthcare or parental health-seeking behavior. Mongru and colleagues showed that, for one of the sites included in this study, the distribution of patients across primary and secondary care was similar before and during the first year of the COVID-19 pandemic, suggesting the observed reductions in patient numbers, especially for those with minor injuries and illnesses,

were a true reflection of fewer children in the community in need of urgent and emergency care [72].

We used the cumulative 14-day rate of new cases per 100,000 of the population to identify high-prevalence countries, but indications for SARS-CoV-2 testing and reporting mechanisms differed between countries, and this could have led to under- or overestimation of national prevalence rates. Moreover, national prevalence numbers might wrongly reflect any regional variation, but, for example, in the UK, identical patterns in ED attendances were seen across the 5 sites, despite large variations in SARS-CoV-2 prevalence during the first phase of the COVID-19 pandemic [73]. Due to the rapid escalation and near-universal introduction of infection prevention measures in European countries during the study period, we were not able determine the role of each of the individual measures on reducing ED attendances.

## Conclusions

Reductions in overall ED attendances were seen across our study sites during the first phase of the COVID-19 pandemic, with health systems across Europe affected similarly. In most sites, there was no suggestion of disproportionate numbers of more severely unwell children. In the first phase of the COVID-19 pandemic, the relative increase in cases of diabetic ketoacidosis or mental health issues might have contributed to a biased perception about increased occurrence, yet this is not supported by an increase in absolute numbers of cases in our data. Our study informs how pediatric emergency medicine can prepare for future pandemics, taking into account that different infectious diseases outbreaks can affect children differently, and illustrates the potential of electronic health records to monitor trends in urgent and emergency care for children.

## Supporting information

**S1 Appendix. Membership of the EPISODES study group.**
(PDF)

**S1 File. EPISODES study protocol.**
(PDF)

**S2 File. EPISODES study registration.**
(PDF)

**S3 File. Clinical report form.**
(PDF)

**S1 Checklist. RECORD checklist.** The RECORD statement—checklist of items, extended from the STROBE statement, which should be reported in observational studies using routinely collected health data.
(PDF)

**S1 Table. List of time windows for data entry.**
(PDF)

**S2 Table. ICD-10 guidance for coding of diagnosis.**
(PDF)

**S3 Table. List of approvals.**
(PDF)

**S4 Table. Overview of participating study sites.**
(PDF)

**S5 Table. List of national social distancing measures.**
(PDF)

**S6 Table. List of national SARS-CoV-2 rates.** The dates and numbers of SARS-CoV02 infections in each of the study sites' countries participating in the EPISODES study.
(PDF)

**S7 Table. Sensitivity analyses: Poisson regression models for ED attendances without TUR003.**
(PDF)

**S1 Fig. Spiderplot for availability of data.**
(PDF)

**S2 Fig. Map of European with all participating study sites.** All participating study sites are highlighted with their study site ID; represented countries in red.
(PDF)

**S3 Fig. Annual ED attendance numbers in 2018 and 2019.** The total number of pediatric ED attendances for each of the study sites for 2018 and 2019 (pre-COVID-19). Overall, the numbers of ED attendances in 2018 and 2019 were similar for each of the study sites, with considerable diversity between the study sites and TUR003 seeing more children in their ED than the other study sites.
(PDF)

**S4 Fig. Annual ED attendance numbers 2018 to 2020 for all sites separately.** The total number of pediatric ED attendances for each of the study sites for the entire study duration (January 2018–May 2020). The y-axis is depicted in log scale.
(PDF)

**S5 Fig. Observed versus predicted ED attendances (%) for different age categories.** The observed versus predicted number of children presenting to EDs in countries across Europe in the weeks following February 2, 2020 until May 11, 2020, for all sites combined, for children (a) aged 0–1 years; (b) 1–2 years; (c) 2–5 years; (d) 5–12 years; and (e) 12–18 years. The color and the size of the dots reflect the actual number of ED attendances for each site and for each time window. The line connects the mean of the observed vs. predicted point estimates for each of the individual sites for each time window.
(PDF)

**S6 Fig. Observed versus predicted ED attendances (%) for different age categories, for individual sites.**
(PDF)

**S7 Fig. Observed versus predicted ED attendances (%) for each country.** The observed versus predicted number of children presenting to EDs in countries across Europe for which data from only 1 study site were available in the weeks following February 2, 2020 until May 11, 2020. A timeline is plotted (dashed line) to show the dates of the introduction of national social distancing measures.[20] One site from the Netherlands and 1 site from Hungary were excluded from these analyses as these sites could not provide data for the entire study duration.
(PDF)

**S8 Fig. Observed versus predicted triage categories (%).** The observed versus predicted number of children presenting to EDs in countries across Europe in the weeks following February 2, 2020 until May 11, 2020, for all sites combined, for children (a) nonurgent and standard

triage classification; (b) urgent triage classification; (c) emergency and very urgent triage classification. The color and the size of the dots reflect the actual number of ED attendances for each site and for each time window. The line connects the mean of the observed vs. predicted point estimates for each of the individual sites for each time window. UK001 did not use a triage system with the emergency and very urgent triage category.
(PDF)

**S9 Fig. Percentage of children admitted to hospital for individual sites.** Percentages of total ED attendances (left) and absolute numbers (right) of children admitted to hospital (top) and PICUs (bottom); comparing the 28-day standardized numbers for the months of January–April for 2018 vs. 2019 vs. 2020. ED, emergency department; PICU, pediatric intensive care unit.
(PDF)

**S10 Fig. Selected clinical diagnoses in the ED for the period January–April over a 3-year period, for high-prevalence countries.** Percentages of total ED attendances (left) and absolute numbers (right) of children with diagnosis of (a) tonsillitis; (b) otitis media; (c) LRTIs; (d) GI infections; (e) appendicitis; (f) testicular torsion; (g) intussusception; (h) mental health issues; (i) diabetic ketoacidosis; (j) radius fracture; and (k) minor head injury; comparing the 28-day standardized numbers for the months of January–April for 2018 vs. 2019 vs. 2020, shown for countries with of a cumulative 14-day rate of new SARS-CoV-2 cases per 100,000 of 80 or more. ED, emergency department; GI, gastrointestinal; LRTI, lower respiratory tract infection; SARS-CoV-2, Severe Acute Respiratory Syndrome Coronavirus 2.
(PDF)

## Acknowledgments

We would like to thank and acknowledge the support and their contributions to the EPI-SODES study of the following persons: Celia Nekrouf (Pediatric Emergency Department, Hopital Universitaire Robert-Debre, Paris, France); Marcello Covino (Emergency Department, Fondazione Policlinico Universitario A. Gemelli IRCCS, Rome, Italy); Benita Lund (Medical Secretary; Pediatric emergency department, Sachs' Children and Youth Hospital, Stockholm, Sweden); Izabella Lottiger (Pediatric Emergency department, Astrid Lindgren Children Hospital, Stockholm, Sweden); Sharna Crosdale (Contracts office; Imperial College London, UK); Sarah Sheedy (Research nurse; Emergency Department, Bristol Royal Hospital for Children, Bristol, UK); James Allbones (Information & Performance Analyst; Birmingham Women's and Children's NHS Foundation Trust, UK), William Jones (University Hospitals of Leicester NHS Trust, UK); Frazer Snowdon and Matthew Ryan (Pediatric emergency department, Alder Hey Children's NHS Foundation Trust, Liverpool, UK), Carlos Saiz-Hernando (IT analyst; Department of Medical Documentation, Cruces University Hospital, Bilbao, Spain); Ellen Barry (research nurse; National Children's Research Centre, Dublin, Ireland) and Fiona Leonard (data analyst, Children's Health Ireland, Ireland); Ernst Eigenbauer (data analyst) and Katharina Lieb (medical student; Department of Pediatrics and Adolescent Medicine, Vienna, Austria); Sanne Vrijland (medical student; Erasmus MC–Sophia Children's hospital, Rotterdam, The Netherlands); Catarina Cordeiro (Pediatric Emergency Service, Hospital Pediátrico, Centro Hospitalar e Universitário de Coimbra, Portugal); Mark Camenzuli (Senior systems administrator; Mater Dei Hospital, Malta); Sandra Distefano (Clinical Performance Unit, Mater Dei Hospital, Malta); Karin Kittl-Mitteregger (HIS Management and Clinical Processes, Paracelsus Medical University, Salzburg, Austria); Marta Arpone (Research Assistant, University of Padova, Italy)

## Author Contributions

**Conceptualization:** Ruud G. Nijman, Kate Honeyford, Ruth Farrugia, Zsolt Bognar, Danilo Buonsenso, Liviana Da Dalt, Tisham De, Ian K. Maconochie, Niccolo Parri, Damian Roland, Camille Aupiais, Michael Barrett, Romain Basmaci, Dorine Borensztajn, Susana Castanhinha, Corinne Vasilico, Sheena Durnin, Paddy Fitzpatrick, Laszlo Fodor, Borja Gomez, Susanne Greber-Platzer, Romain Guedj, Stuart Hartshorn, Florian Hey, Lina Jankauskaite, Daniela Kohlfuerst, Mojca Kolnik, Mark D. Lyttle, Patrícia Mação, Maria Inês Mascarenhas, Shrouk Messahel, Esra Akyüz Özkan, Zanda Pučuka, Sofia Reis, Alexis Rybak, Malin Ryd Rinder, Ozlem Teksam, Caner Turan, Valtýr Stefánsson Thors, Roberto Velasco, Silvia Bressan, Henriette A. Moll, Rianne Oostenbrink, Luigi Titomanlio.

**Data curation:** Ruud G. Nijman, Kate Honeyford, Ruth Farrugia, Katy Rose, Zsolt Bognar, Danilo Buonsenso, Liviana Da Dalt, Tisham De, Ian K. Maconochie, Niccolo Parri, Damian Roland, Tobias Alfven, Camille Aupiais, Michael Barrett, Romain Basmaci, Dorine Borensztajn, Susana Castanhinha, Corinne Vasilico, Sheena Durnin, Paddy Fitzpatrick, Laszlo Fodor, Borja Gomez, Susanne Greber-Platzer, Romain Guedj, Stuart Hartshorn, Florian Hey, Lina Jankauskaite, Daniela Kohlfuerst, Mojca Kolnik, Mark D. Lyttle, Patrícia Mação, Maria Inês Mascarenhas, Shrouk Messahel, Esra Akyüz Özkan, Zanda Pučuka, Sofia Reis, Alexis Rybak, Malin Ryd Rinder, Ozlem Teksam, Caner Turan, Valtýr Stefánsson Thors, Roberto Velasco, Silvia Bressan, Henriette A. Moll, Rianne Oostenbrink, Luigi Titomanlio.

**Formal analysis:** Ruud G. Nijman, Kate Honeyford, Tisham De, Damian Roland, Camille Aupiais, Michael Barrett, Romain Basmaci, Dorine Borensztajn, Susana Castanhinha, Corinne Vasilico, Sheena Durnin, Paddy Fitzpatrick, Laszlo Fodor, Borja Gomez, Susanne Greber-Platzer, Romain Guedj, Stuart Hartshorn, Florian Hey, Lina Jankauskaite, Daniela Kohlfuerst, Mojca Kolnik, Mark D. Lyttle, Patrícia Mação, Maria Inês Mascarenhas, Shrouk Messahel, Esra Akyüz Özkan, Zanda Pučuka, Sofia Reis, Alexis Rybak, Malin Ryd Rinder, Ozlem Teksam, Caner Turan, Valtýr Stefánsson Thors, Roberto Velasco, Silvia Bressan, Henriette A. Moll, Rianne Oostenbrink, Luigi Titomanlio.

**Funding acquisition:** Ruud G. Nijman, Ian K. Maconochie, Damian Roland, Henriette A. Moll, Rianne Oostenbrink, Luigi Titomanlio.

**Investigation:** Ruud G. Nijman, Kate Honeyford, Ruth Farrugia, Zsolt Bognar, Danilo Buonsenso, Liviana Da Dalt, Tisham De, Ian K. Maconochie, Niccolo Parri, Damian Roland, Tobias Alfven, Michael Barrett, Romain Basmaci, Dorine Borensztajn, Susana Castanhinha, Corinne Vasilico, Sheena Durnin, Paddy Fitzpatrick, Laszlo Fodor, Borja Gomez, Susanne Greber-Platzer, Romain Guedj, Stuart Hartshorn, Florian Hey, Lina Jankauskaite, Daniela Kohlfuerst, Mojca Kolnik, Mark D. Lyttle, Patrícia Mação, Maria Inês Mascarenhas, Shrouk Messahel, Esra Akyüz Özkan, Zanda Pučuka, Sofia Reis, Alexis Rybak, Malin Ryd Rinder, Ozlem Teksam, Caner Turan, Valtýr Stefánsson Thors, Roberto Velasco, Silvia Bressan, Henriette A. Moll, Rianne Oostenbrink, Luigi Titomanlio.

**Methodology:** Ruud G. Nijman, Kate Honeyford, Ruth Farrugia, Katy Rose, Zsolt Bognar, Danilo Buonsenso, Liviana Da Dalt, Tisham De, Ian K. Maconochie, Niccolo Parri, Damian Roland, Tobias Alfven, Camille Aupiais, Michael Barrett, Romain Basmaci, Dorine Borensztajn, Susana Castanhinha, Corinne Vasilico, Sheena Durnin, Paddy Fitzpatrick, Laszlo Fodor, Borja Gomez, Susanne Greber-Platzer, Romain Guedj, Stuart Hartshorn, Florian Hey, Lina Jankauskaite, Daniela Kohlfuerst, Mojca Kolnik, Mark D. Lyttle, Patrícia Mação, Maria Inês Mascarenhas, Shrouk Messahel, Esra Akyüz Özkan, Zanda Pučuka,

Sofia Reis, Alexis Rybak, Malin Ryd Rinder, Ozlem Teksam, Caner Turan, Valtýr
Stefánsson Thors, Roberto Velasco, Silvia Bressan, Henriette A. Moll, Rianne Oostenbrink,
Luigi Titomanlio.

**Project administration:** Ruth Farrugia, Katy Rose, Zsolt Bognar, Danilo Buonsenso, Liviana
Da Dalt, Tisham De, Ian K. Maconochie, Niccolo Parri, Damian Roland, Tobias Alfven,
Camille Aupiais, Michael Barrett, Romain Basmaci, Dorine Borensztajn, Susana
Castanhinha, Corinne Vasilico, Sheena Durnin, Paddy Fitzpatrick, Laszlo Fodor, Borja
Gomez, Susanne Greber-Platzer, Romain Guedj, Stuart Hartshorn, Florian Hey, Lina
Jankauskaite, Daniela Kohlfuerst, Mojca Kolnik, Mark D. Lyttle, Patrícia Mação, Maria Inês
Mascarenhas, Shrouk Messahel, Esra Akyüz Özkan, Zanda Pučuka, Sofia Reis, Alexis
Rybak, Malin Ryd Rinder, Ozlem Teksam, Caner Turan, Valtýr Stefánsson Thors, Roberto
Velasco, Silvia Bressan, Henriette A. Moll, Rianne Oostenbrink, Luigi Titomanlio.

**Resources:** Ruud G. Nijman, Kate Honeyford, Liviana Da Dalt, Tisham De, Ian K.
Maconochie, Damian Roland, Tobias Alfven, Camille Aupiais, Michael Barrett, Romain
Basmaci, Dorine Borensztajn, Susana Castanhinha, Corinne Vasilico, Sheena Durnin,
Paddy Fitzpatrick, Laszlo Fodor, Borja Gomez, Susanne Greber-Platzer, Romain Guedj,
Stuart Hartshorn, Florian Hey, Lina Jankauskaite, Daniela Kohlfuerst, Mojca Kolnik, Mark
D. Lyttle, Patrícia Mação, Maria Inês Mascarenhas, Shrouk Messahel, Esra Akyüz Özkan,
Zanda Pučuka, Sofia Reis, Alexis Rybak, Malin Ryd Rinder, Ozlem Teksam, Caner Turan,
Valtýr Stefánsson Thors, Roberto Velasco, Silvia Bressan, Henriette A. Moll, Rianne
Oostenbrink, Luigi Titomanlio.

**Supervision:** Ruud G. Nijman, Kate Honeyford, Zsolt Bognar, Danilo Buonsenso, Liviana Da
Dalt, Niccolo Parri, Damian Roland, Tobias Alfven, Camille Aupiais, Michael Barrett,
Romain Basmaci, Dorine Borensztajn, Susana Castanhinha, Corinne Vasilico, Sheena
Durnin, Paddy Fitzpatrick, Laszlo Fodor, Borja Gomez, Susanne Greber-Platzer, Romain
Guedj, Florian Hey, Lina Jankauskaite, Daniela Kohlfuerst, Mojca Kolnik, Mark D. Lyttle,
Patrícia Mação, Maria Inês Mascarenhas, Shrouk Messahel, Zanda Pučuka, Sofia Reis,
Alexis Rybak, Malin Ryd Rinder, Silvia Bressan, Henriette A. Moll, Rianne Oostenbrink,
Luigi Titomanlio.

**Validation:** Kate Honeyford, Ruth Farrugia, Katy Rose, Zsolt Bognar, Danilo Buonsenso, Ian
K. Maconochie, Esra Akyüz Özkan, Ozlem Teksam, Caner Turan, Valtýr Stefánsson Thors.

**Visualization:** Ruud G. Nijman, Ruth Farrugia, Katy Rose, Liviana Da Dalt, Stuart Hartshorn,
Roberto Velasco, Rianne Oostenbrink, Luigi Titomanlio.

**Writing – original draft:** Ruud G. Nijman, Kate Honeyford, Ruth Farrugia, Katy Rose, Ian K.
Maconochie, Rianne Oostenbrink, Luigi Titomanlio.

**Writing – review & editing:** Ruud G. Nijman, Kate Honeyford, Ruth Farrugia, Katy Rose,
Zsolt Bognar, Danilo Buonsenso, Liviana Da Dalt, Tisham De, Ian K. Maconochie, Niccolo
Parri, Damian Roland, Tobias Alfven, Camille Aupiais, Michael Barrett, Romain Basmaci,
Dorine Borensztajn, Susana Castanhinha, Corinne Vasilico, Sheena Durnin, Paddy
Fitzpatrick, Laszlo Fodor, Borja Gomez, Susanne Greber-Platzer, Romain Guedj, Stuart
Hartshorn, Florian Hey, Lina Jankauskaite, Daniela Kohlfuerst, Mojca Kolnik, Mark D.
Lyttle, Patrícia Mação, Maria Inês Mascarenhas, Shrouk Messahel, Esra Akyüz Özkan,
Zanda Pučuka, Sofia Reis, Alexis Rybak, Malin Ryd Rinder, Ozlem Teksam, Caner Turan,
Valtýr Stefánsson Thors, Roberto Velasco, Silvia Bressan, Henriette A. Moll, Rianne
Oostenbrink, Luigi Titomanlio.

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
