## [Editor Report · Decision Letter 0]

22 Mar 2022

Dear Dr Nijman, 

Thank you for submitting your manuscript entitled "Patterns of presentations of children to emergency departments across Europe and the impact of the COVID-19 pandemic: retrospective observational multinational study." for consideration by PLOS Medicine.

Your manuscript has now been evaluated by the PLOS Medicine editorial staff and I am writing to let you know that we would like to send your submission out for external peer review.

Please re-submit your manuscript within two working days, i.e. by Mar 24 2022 11:59PM.

Kind regards,

Callam Davidson

Associate Editor

PLOS Medicine

---

## [Decision Letter · Decision Letter 1]

11 May 2022

Dear Dr. Nijman,

Thank you very much for submitting your manuscript "Patterns of presentations of children to emergency departments across Europe and the impact of the COVID-19 pandemic: retrospective observational multinational study." (PMEDICINE-D-22-00899R1) for consideration at PLOS Medicine. 

Your paper was evaluated by an associate editor and discussed among all the editors here. It was also discussed with an academic editor with relevant expertise, and sent to independent reviewers, including a statistical reviewer. The reviews are appended at the bottom of this email and any accompanying reviewer attachments can be seen via the link below:

[LINK]

In light of these reviews, I am afraid that we will not be able to accept the manuscript for publication in the journal in its current form, but we would like to consider a revised version that addresses the reviewers' and editors' comments. Obviously we cannot make any decision about publication until we have seen the revised manuscript and your response, and we plan to seek re-review by one or more of the reviewers. 

We hope to receive your revised manuscript by Jun 01 2022 11:59PM. Please email us (plosmedicine@plos.org) if you have any questions or concerns.

We look forward to receiving your revised manuscript. 

Sincerely,

Callam Davidson, 

PLOS Medicine

plosmedicine.org

Please revise your title. "Impact of" should be used only if causality can be inferred, i.e., for an RCT. Please also remove the word ‘retrospective’. I suggest ‘Presentation of children to emergency departments across Europe and the COVID-19 pandemic: a multinational observational study’, or similar.

Abstract: 

* Please combine the Methods and Findings sections into one section, “Methods and findings”.

* Please change ‘Interpretation’ to ‘Conclusions’

* Please remove funding information from the abstract and instead include this in your Financial Disclosure (submission form).

* Please remove the trial registry information and instead include this in your Methods.

Abstract Background: Provide the context of why the study is important. The final sentence should clearly state the study question.

Abstract Methods and Findings:

* Please include p values when quantifying the main results.

* Please include the important variables that are adjusted for in the analyses.

Please update your line numbering to be continuous (rather than restarting on each new page).

Citations should precede punctuation.

Please confirm that individual patient informed consent was waived by all participating IRBs.

Please remove the Role of the Funding Source section. 

Please ensure that the study is reported according to the STROBE guideline, and include the completed STROBE checklist as Supporting Information. Please add the following statement, or similar, to the Methods: "This study is reported as per the Strengthening the Reporting of Observational Studies in Epidemiology (STROBE) guideline (S1 Checklist)."

Did your study have a prospective protocol or analysis plan? Please state this (either way) early in the Methods section.

Please do not cite figures that present results in the Methods section.

The shading in Table 1 does not appear to be consistent.

Please update the title of Figure 2 to ‘Observed versus predicted emergency department attendances (%)’ or similar. Please update the titles of Figure 3 and 6 in a similar way.

Figures 2, 3, and 6: You refer to ‘predicted’ in the title and legend but ‘expected’ in the axis label, please update to be consistent.

Table 2: 

* Please report P<0.001 rather than P<0.0001.

* Please specify the variables controlled for in the legend.

* Please present the unadjusted comparisons as well as the adjusted comparisons.

Abstract/Results/Discussion: Your study is observational and therefore causality cannot be inferred. Please remove language that implies causality, such as ‘impact’. Refer to associations instead.

Figure 5 is very large – please consider reducing the size of the individual panels (as in Figure 4) or moving this figure to the Supporting Information.

Figure 6:

* The x-axis is labelled incorrectly.

* Please indicate the meaning of the error bars.

Please quantify the findings presented in the Sensitivity analyses subsection, with 95% CI and p-values where appropriate, rather than describing a result as less significant.

Discussion: 

* Please temper claims of primacy (such as that found at line 2, page 23) by adding ‘to our knowledge’.

* Figure S17 appears to present UK government data, please consider citing the appropriate data in your References rather than including as an additional figure in the Supporting Information.

Please remove the Data sharing statement, Conflict of interest, Funding source, and Author contributions from the end of the main text and instead report this information in the appropriate parts of the submission form. In the event of publication, this information will be presented as metadata.

For internet sources in the References, please present the date cited.

Comments from the reviewers:

Reviewer #1: This study has investigated the impact of the COVID-19 pandemic and infection prevention measures on children visiting emergency departments across Europe. The study has provided important implications for the understanding of the change of the ED attendances during the pandemic. However, to make the review more valuable, I have some comments and hope these will help to improve this manuscript.

1. Page 1, Line 9: aged <16 years? Why the authors stated that upper age limit varied between sites at between 16 and <18 years old?

2. Page 2-4: Please give the full name for the "COVID-19" and "SARS-CoV-2" when they appeared for the first time.

3. Page 3, Line 4-9: I seem to disagree with the authors. The reason why children less likely to develop symptoms of severe acute SARS-CoV-2 infection may not attributable to reduced numbers of children visiting urgent and emergency care services. The authors should review more relevant studies to clarify the point clearly.

4. Page 3, Line 20: did the possible link exist? How did the authors infer this conclusion?

5. Page 4, Line 1-2: Why did author only compare the situations during the first phase of the COVID-19 pandemic with the two previous years? The pandemic has last for more than two years. The authors should consider a longer period if possible.

6. Page 5, Line 5: what were the triage urgency levels? How did the authors define the different levels? 

7. Page 6, Line 1: How did the author adjust short-term monthly forecasts to weekly estimates of predicted numbers?

8. Page 8, Line 17: "Description of infection prevention measures and SARS-CoV-2 prevalence". The authors should combine the two aspects of information together in this descriptive paragraph. 

9. Page 10, Line 2-3: How did the authors define the organisational COVID-19 measures? Please clarify the point in the method section.

10. Page 19, Line 21-23; Page 20, Line 1-3: The authors should discuss more about multifactorial reasons. 

11. Fig 1 should be revised. There are many overlaps among the figure legends which make it hard to get important information.

12. Could authors provide a map about the distributions about the site? The map will help to the understanding about the information in the Figure 2 and Figure 3.

Reviewer #2: "Patterns of presentations of children to emergency departments (ED) across Europe and the impact of the COVID-19 pandemic: retrospective observational multinational study" examines retrospective health data from 16 European countries from Jan 2018 to May 2020, to attempt to quantify the impact of the coronavirus pandemic on ED presentations by children aged <16 years. Data prior to the pandemic was used to evaluate that from after the pandemic, after trend and seasonal adjustments. The main findings are that a reduction in ED attendances, hospital admissions and high triage urgencies was observed after the start of the pandemic, broadly in line with prior literature.

The analysis appears largely comprehensive and sound. However, some issues might be considered:

1. The relatively short period of assessment (Feb 2020 to May 2020) may be noted, despite it currently being April 2022. Might it have been possible to describe a longer period post-pandemic start, especially since that might allow potential annual cycles/patterns to be analyzed?

2. In the Abstract, it is claimed that "...ED attendances were relatively higher in ...children aged >12 months (12-<24 months IRR 0.89, 95% CI 0.86 to 0.92; 2-<5years IRR 0.84, 95% CI 0.82 to 0.87; 5-<12 years IRR 0.74, 95% CI 0.72 to 0.76; 12-<16 years IRR 0.74, 95% CI 0.71 to 0.77; vs. age <12 months as reference group)". However, it appears that attendances were actually relatively lower (IRR<1) in older children compared to age <12 months. This might be checked.

3. In Page 4, it is stated that the study included 38 sites from 16 European countries. The representativeness of the sites might be commented upon. Analysis for geographic status (urban vs. rural) might be considered.

4. In Page 4, it is stated that data from before Feb 2020 was collected monthly, and weekly after Feb 2020 (after the start of the pandemic). Was this choice of period (monthly vs. weekly) due to the limitations of the data on the online platform for the period before Feb 2020, or was it a conscious choice?

5. In Page 4, it is stated that standardized (aggregated) data was used. The next page then describes triage urgency levels that were categorized in three categories. For these relevant data/variables, they might be listed (possibly in supplementary material), and clarified as to whether all participating EDs followed the same classifications/system. If not, the conversions to a standardized norm might be given.

6. In Page 5, it is stated that "For diagnosis coding, ICD-10 codes were issued for guidance (S5 Table), but an internally and temporally consistent coding approach was encouraged for each of the individual sites, acknowledging different coding systems and strategies in the ED". This might be further clarified. In particular, while the approach was encouraged, was it actually implemented for all sites? Given the different coding systems/strategies, how were they standardized?

7. In Page 5, it is stated that Holt-Winters exponential smoothing was used to make short-term forecasts for February to May 2020. A brief analysis of the chosen smoothing method on known data (e.g. fitting with data from Jan 2018 to Jan 2019, predicting and comparing against Feb to May 2019) might be strongly considered.

8. In Page 14, Table 2 contains analysis for low SARS-CoV-2 prevalence. Analysis for more stratification (e.g. percentile groups) might be considered.

9. In Page 15, while it is claimed that no increase in number of deaths in ED was observed, the IRR from Table 2 appears to be 1.75 (0.88-3.07). This appears to actually suggest a considerable increase, if not statistically-significant. As such, this point might be discussed further.

10. In Page 16, the selection criteria for the various diagnoses presented in Table 3 might be briefly discussed.

11. In Figure 4, deaths in the ED were described. It might be considered to also show figures for (excess) deaths for the age group of interest (<16 years), for context.

12. Related to the above, medical options for children during the studied post-pandemic start period might be briefly discussed. Were they possibly diverted to private clinics/home care/telemedicine?

13. Any patterns from country-level analyses (as from Table/Figure 1) relating response measures to general outcomes might be discussed in the main text.

14. For Table/Figure 1, the source and reliability of data relating to (cumulative) rate of new SARS-CoV-2 cases might be discussed, given the possible difficulty of testing especially earlier in the pandemic.

Reviewer #3: This is an observational study that reports the changes in presentations of children to emergency departments. The authors have collected data across multiple European countries. Documenting this important change in service demand is important for understanding for future pandemic planning, so while the results are not unexpected they are still valuable. 

These comments are intended to improve the usefulness of the manuscript. 

1. The paper is well written and generally accessible, though the presentation of data can be a little confusing. This may be made easier by having an initial figure one that simply reports the total number of attendances and admissions over time. It feels a jump to go straight into actual / predicted

2. The finding about low acuity cases being the group that stayed away most is important. The authors should enhance the discussion about why this has occurred to explicitly consider whether there are any behavioral factors at play. Within adults, there was definitely a fear about attending hospital as a risk of exposure e.g. https://www.cureus.com/articles/34364-the-attend-study-a-retrospective-observational-study-of-emergency-department-attendances-during-the-early-stages-of-the-covid-19-pandemic It is very plausible that parents perception of risk / benefit to attending a paediatric ED with lower acuity presentations might have changed.

[LINK]

---

## [Decision Letter · Decision Letter 2]

18 Jul 2022

Dear Dr. Nijman,

Thank you very much for re-submitting your manuscript "Presentations of children to emergency departments across Europe and the COVID-19 pandemic: a multinational observational study." (PMEDICINE-D-22-00899R2) for review by PLOS Medicine.

I have discussed the paper with my colleagues and the academic editor and it was also seen again by three reviewers. I am pleased to say that provided the remaining editorial and production issues are dealt with we are planning to accept the paper for publication in the journal.

[LINK]

We look forward to receiving the revised manuscript by Jul 25 2022 11:59PM.   

Sincerely,

Callam Davidson, 

Associate Editor 

PLOS Medicine

plosmedicine.org

Requests from Editors:

Please provide the URL from which data are available in your Data Availability Statement. 

Please provide the full study protocol in your Supporting Information and cite this in your Methods.

Please ensure that all numbers presented in the abstract are present and identical to numbers presented in the main manuscript text.

Line 27 and line 304: Please quantify the association with general admissions.

Line 56-57: Please quantify the key results presented in this bullet point (with 95% CI and p-values).

Please aim to trim your author summary such that there are 2-3 bullet points per question.

Your study is observational and therefore causality cannot be inferred. Please remove language that implies causality, such as that found at lines 63-64. Refer to associations instead.

My sincere apologies but in my previous decision letter I mistakenly suggested the STROBE checklist ought to be included in your Supporting Information. This ought to have been the RECORD checklist (https://www.record-statement.org/checklist.php). Please update the statement at lines 110-111 and include this checklist in place of the STROBE (again, referring to paragraph number and section headings rather than page numbers).

S8 Figure: Please confirm that the appropriate usage rights apply to the use of this map. Please see our guidelines for map images: https://journals.plos.org/plosmedicine/s/figures#loc-maps

Figure S17: AUS001 is missing.

Line 370: Please present the results of the sensitivity analyses in full in the Supporting Information.

Comments from Reviewers:

Reviewer #1: I have read the revised manuscript "Presentations of children to emergency departments across Europe and the COVID-19 pandemic: a multinational observational study". The authors have made revisions well, point by point. However, minor revisions should be made to improve the manuscript for some minor errors and the syntax and language.

For example: 

1、 Line 51: "…16 years and under…" should be "…18 years and under…"?

2、 Improper use of commas in the Abstract of "Concerns were raised about the potential for delayed, and more severe, presentations, and an increase in diagnoses such as diabetic keto-acidosis and mental health issues." 

3、 Abstract: "This observational multinational study …" should be "This multinational observational study…"

4、 Line 109: "Upper age limit…" should be "The upper age limit…"

Reviewer #2: We thank the authors for addressing our previous comments. Brief mention might be made about (over-)stratification in the manuscript if considered appropriate.

Reviewer #3: The authors have done a diligent job of the revisions and answered the questions adequately

[LINK]

---

## [Editor Report · Decision Letter 3]

25 Jul 2022

Dear Dr. Nijman,

Thank you very much for re-submitting your manuscript "Presentations of children to emergency departments across Europe and the COVID-19 pandemic: a multinational observational study." (PMEDICINE-D-22-00899R3) for review by PLOS Medicine.

The remaining issues that need to be addressed are listed at the end of this email. 

In revising the manuscript for further consideration here, please ensure you address the specific points made by the editors. In your rebuttal letter you should indicate your response to the reviewers' and editors' comments and the changes you have made in the manuscript. Please submit a clean version of the paper as the main article file. A version with changes marked must also be uploaded as a marked up manuscript file.

We hope to receive your revised manuscript within 3 days. Please email us (plosmedicine@plos.org) if you have any questions or concerns.

We look forward to receiving the revised manuscript by Aug 04 2022 11:59PM.   

Sincerely,

Callam Davidson, 

Associate Editor 

PLOS Medicine

plosmedicine.org

Requests from Editors:

Line 106: Please add ‘…and the study protocol is available in the Supporting Information (S1 File). 

Line 58: Please add ‘The findings suggest that…’.

Line 60: Please add ‘…of infection prevention measures…’.

S21 Table: Please add the corresponding footnote for the flag (^), which appears to be missing.

Your response to Reviewer #2’s previous comment does not address (or respond to) their request ‘Brief mention might be made about (over-)stratification in the manuscript if considered appropriate.’ Please either include this or provide a response to explain why you have chosen not to include this. 

S3 checklist title/legend: Please update this to ‘RECORD’ rather than ‘REPORT’ checklist. 

Lines 368-370: Please update to ‘There was also a change between PICU admissions (IRR 1.13, 95% CI 1.00 – 1.28, p=0.045) and admissions in general. Though the comparison remained statistically significant, the association was weaker.’

---

## [Editor Report · Decision Letter 4]

28 Jul 2022

Dear Dr Nijman, 

On behalf of my colleagues and the Academic Editor, Dr Zulfiqar Bhutta, I am pleased to inform you that we have agreed to publish your manuscript "Presentations of children to emergency departments across Europe and the COVID-19 pandemic: a multinational observational study." (PMEDICINE-D-22-00899R4) in PLOS Medicine.

When making the formatting changes please also update your Data Availability Statement to include the full doi URL at which your study data can be found (i.e., https://doi.org/10.14469/hpc/10685).

PRESS

Sincerely, 

Callam Davidson 

Associate Editor 

PLOS Medicine